# Transformers through the lens of support-preserving maps between measures

## Abstract

Transformers are deep architectures that define "in-context maps" which enable predicting new tokens based on a given set of tokens (such as a prompt in NLP applications or a set of patches for a vision transformer). Previous work has studied the ability of these architectures to handle an arbitrarily large number of context tokens. To mathematically, uniformly analyze their expressivity, previous work considered the case that the mappings are conditioned on a context represented by a probability distribution which becomes discrete for a finite number of tokens. Modeling neural networks as maps on probability measures has multiple applications, such as studying Wasserstein regularity, proving generalization bounds and doing a mean-field limit analysis of the dynamics of interacting particles as they go through the network. In this work, we study the question what kind of maps between measures are transformers. We fully characterize the properties of maps between measures that enable these to be represented in terms of in-context maps via a push forward. On the one hand, these include transformers; on the other hand, transformers universally approximate representations with any continuous in-context map. These properties are preserving the cardinality of support and that the regular part of their Fréchet derivative is uniformly continuous. Moreover, we show that the solution map of the Vlasov equation, which is of nonlocal transport type, for interacting particle systems in the mean-field regime for the Cauchy problem satisfies the conditions on the one hand and, hence, can be approximated by a transformer; on the other hand, we prove that the measure-theoretic self-attention has the properties that ensure that the infinite depth, mean-field measure-theoretic transformer can be identified with a Vlasov flow.

## 1 Introduction

Transformers have revolutionized the field of machine learning with their powerful attention mechanisms as introduced by Vaswani et al. (2017). The exceptional performance and expressivity of large-scale transformers have been empirically well established for both NLP (Brown et al. (2020)) and vision applications (Dosovitskiy et al. (2020)). One key property of these architectures is their ability to leverage contexts of arbitrary length, which enables the parameterization of "in-context" mappings with an arbitrarily large complexity. The previous work (Furuya et al. (2024)) studied this by analyzing the expressivity of mappings that are conditioned on a context represented by a probability distribution of tokens which becomes discrete for a finite number of these. By implication, transformers are viewed as maps between measures. Here, we present a full characterization of maps between measures that can be represented by these measure-theoretic transformers, that is, we address the question which class of mappings between measures can be identified with transformers.

**Mathematical modeling of transformers.** It is now customary to describe transformers as performing "in context" prediction, which means that it maps token to token, while this map depends on a set of previously seen tokens. The size of this context might be very long, possibly arbitrarily long, which has been addressed in Furuya et al. (2024) that concerns the transformers as universal in-context learners. The ability of trained transformers to effectively perform in-context computation has been supported by both empirical studies (von Oswald et al. (2023)) and theoretical ones (Ahn et al. (2024); Mahankali et al. (2023); Sander et al. (2024); Zhang et al. (2023)) on simplified

architectures (typically with linear attention) and specific data generation processes. The connection between transformers and graph neural networks is exposed in Müller et al. (2023).

It has been noted that in order to make a comprehensive analysis of arbitrarily long token lengths, and to describe a "mean-field" limit of an infinite number of tokens, it is natural to view attention as operating over probability distributions of tokens (Vuckovic et al. (2020); Sander et al. (2022)). The regularity (Lipschitz continuity) of the resulting attention layers was analyzed in Castin et al. (2024).

Deep transformers (with residual or skip connections) have been described by a coupled system of particles evolving across the layers. Such systems are fundamental in modeling phenomena across physics, biology, and engineering. This connection has been exploited by Geshkovski et al. (2024) who studied measure-to-measure interpolation using transformers. The analysis of the clustering properties of such an evolution was studied in Geshkovski et al. (2023a;b). Biswal et al. (2024) further investigate the use of transformers to approximate the mean-field dynamics of interacting particle systems exhibiting collective behavior. They establish theoretical bounds on the distance between the true mean-field dynamics and those obtained using a transformer, by lifting it from a sequence-to-sequence map to a map on measures upon taking the expectation of a finite-dimensional transformer with respect to a product measure. From a different viewpoint, this connection will be further developed here, in general for mappings between measures satisfying the conditions to be representable by measure-theoretic transformers. The structure of the interacting particle system enables concrete connections to established mathematical topics, including nonlinear, nonlocal transport equations, Wasserstein gradient flows, and collective behavior models.

**Universality of transformers.**   Yun et al. (2019) provides, to the best of our knowledge, the most detailed account of the universality of transformers. The authors rely on shallow transformers with only two heads and require that the transformers operate over an embedding dimension which grows with the number of tokens. This result is refined in Nath et al. (2024) and emphasizes the difficulty of attention mechanisms to capture smooth functions.

We note that there exist variations of the original transformer's architecture which enjoy universality results, for instance, the Sumformer (Alberti et al. (2023)) and stochastic deep network (De Bie et al. (2019)); these also require an embedding dimension that grows with the number of tokens. We furthermore mention the introduction of probabilistic transformers (Kratsios et al. (2023)) which can approximate embeddings of metric spaces. The work of Agrachev and Letrouit (2024) provides an abstract universal interpolation result for equivariant architectures under genericity conditions; however, it is not known whether there exist generic attention maps.

While this is not directly related to the analysis presented here, some works study the expressivity of transformers when operating on a discrete set of tokens as formal systems (Chiang et al. (2023); Merrill and Sabharwal (2023); Strobl et al. (2024); Elhage et al. (2021)). Another line of work studies the impact of positional encoding on their expressivity (Luo et al. (2022)).

Furuya et al. (2024) provide a rigorous formalization of transformer expressivity and continuity as operating over the space of probability distributions through its in-context mapping. The main mathematical result is the universal approximation of in-context mappings for the unmasked and the masked settings, considering deep transformers with a fixed embedding dimension, but which are universal for an arbitrary number of tokens. A more constructive approach, although applicable to a narrower class of functions, is proposed by Wang et al. (2024). Sander and Peyré (2024) introduce a framework to analyze the expressivity of deep transformers in next-token prediction, while exploring how successive attention layers solve a causal kernel least squares regression problem to predict the next token accurately.

## 1.1 OUR CONTRIBUTIONS

The central question posed here, is whether a support-preserving map between measures can be characterized as the push forward with an in-context map or not. We answer this question in the affirmative by introducing a "certain" smoothness condition, which roughly entails that a "certain" derivative of the map is uniformly continuous. We provide a counterexample, showing that this condition is essential. Our proof is essentially constructive.

Applying this result and the underlying analysis, we prove that measure-theoretic transformers approximate such support-preserving maps, using the results of Furuya et al. (2024). This settles the full characterization of measure-theoretic transformers.

Finally, we show that the solution operator of the Vlasov equation, which is of nonlocal transport type, for the Cauchy problem satisfies the above mentioned condition(s) as a map between initial and final measures. This provides a bridge between interacting particle systems, in the mean-field regime, in the general context of measure-theoretic transformers. (A second-order generalization of measure-theoretic transformers yields a similar result for the solution operator of the kinetic Cucker-Smale equation (Biswal et al. (2024)).)

We first present the analysis relating support-preserving maps between measures with in-context maps that define measure-theoretic transformers. We later show, in an appendix, that "classical" transformers arise as a limiting case through (sub)sequences of discrete measures determined by tokens. The correspondence with Vlasov flows is established in the mean-field sense and is based on an infinite-depth limit.

## 1.2 NOTATION

Let $\Omega \subset \mathbb{R}^d$ be a compact set. We denote by $\mathcal{P}(\Omega)$ the space of probability measures on $\Omega$. Below, all measures $\mu$ on subset $\Omega$ of $\mathbb{R}^d$ are defined on the $\sigma$-algebra of the Borel sets of $\Omega$. We denote by $C(\Omega)$ the space of continuous functions from $\Omega$ to $\mathbb{R}$, and the dual coupling between $\varphi \in C(\Omega)$ and $\mu \in \mathcal{P}(\Omega)$ by

$$\langle \varphi, \mu \rangle := \int_\Omega \varphi(x) \mathrm{d}\mu(x).$$

We use the notations of Wasserstein distance as $W_p$ for $1 \le p < \infty$. We extend $\mathcal{P}(\Omega)$, that is, the set of all probability measures to the set of all strictly positive, finite measures

$$\mathcal{M}^+(\Omega) = \{s\mu : \ \mu \in \mathcal{P}(\Omega), \ s > 0\}.$$

We also extend the $W_1$ distance to $\mathcal{M}^+(\Omega)$ by defining for $\mu_1, \mu_2 \in \mathcal{P}(\Omega)$ and $s_1, s_2 > 0$

$$W_1(s_1\mu_1, s_2\mu_2) = W_1(\mu_1, \mu_2) + |s_1 - s_2|,$$

see Lombardini and Rossi (2022). We write

$$\mathcal{M}^+_{fin,(n)}(\Omega) := \left\{ \sum_{i=1}^n a_i \delta_{x_i} \in \mathcal{M}^+(\Omega) : \ x_i \in X, \ a_i > 0 \right\},$$

$$\mathcal{M}^+_{fin}(\Omega) := \bigcup_{n=1}^\infty \mathcal{M}^+_{fin,(n)}(\Omega) = \left\{ \sum_{i=1}^n a_i \delta_{x_i} \in \mathcal{M}^+(\Omega) : \ x_i \in \Omega, \ a_i > 0, \ n \in \mathbb{N} \right\}.$$

Finally, we denote by $\mathcal{M}^+_{fin,dif,(n)}(\Omega)$ the measures of the form $\mu = \sum_{i=1}^n a_i \delta_{x_i} \in \mathcal{M}_{fin,(n)}(\Omega)$, where $a_j > 0$ and for all non-empty subsets $J, K \subset \{1, 2, \ldots, n\}$ satisfying $J \cap K = \emptyset$ it holds that

$$\sum_{j \in J} a_j \neq \sum_{k \in K} a_k.$$

We set $\mathcal{M}^+_{fin,dif}(\Omega) = \bigcup_{n=1}^\infty \mathcal{M}^+_{fin,dif,(n)}(\Omega)$. For a continuous map $g : \Omega \to \Omega$ and a measure $\mu$ the push-forward measure of $\mu$ in the map $g$ is the measure $g_\#\mu(A) := \mu(g^{-1}(A))$, where $A \subset \Omega$ is an open set. For further details pertaining to these notions, we refer to Appendix A.1.

We state the following lemma, which is proved in Appendix B.1.

**Lemma 1.** $\mathcal{M}^+_{fin,dif}(\Omega)$ *is dense in* $\mathcal{M}^+(\Omega)$ *in the 1-Wasserstein topology.*

## 2 DEFINITIONS AND PROPERTIES OF THE RELEVANT MAPS

### 2.1 SUPPORT-PRESERVING MAPS AND IN-CONTEXT MAPS

**Definition 1.** *We say that* $f : \mathcal{M}^+(\Omega) \to \mathcal{M}^+(\mathbb{R}^{d'})$ *is a support-preserving map if for all finitely supported measures of the form,*

$$\mu = \sum_{i=1}^n a_i \delta_{x_i} \in \mathcal{M}^+_{fin}(\Omega), \tag{1}$$

*where $a_i = \frac{c}{n}$, $c \in \mathbb{R}_+$, and $x_i \in \Omega$, there exist $y_1, ..., y_n \in \mathbb{R}^{d'}$ such that*

$$f(\mu) = \sum_{i=1}^n a_i \delta_{y_i} \in \mathcal{M}_{fin}^+(\mathbb{R}^{d'}) \tag{2}$$

*and satisfy the condition*

$$if \ x_j = x_i \ then \ y_j = y_i. \tag{3}$$

The consideration of support-preserving maps to study transformers is natural; see Section 4.1 and formula (23) in Appendix A.3 and A.4 for a detailed discussion. Let $(x_1, x_2, \ldots, x_n) \in \Omega^n$ be the sequences of $n$ tokens in $\Omega \subset \mathbb{R}^d$, and let the union of all these be $X_d = \bigcup_{n=1}^\infty \Omega^n$. A sequence $(x_1, x_2, \ldots, x_n)$ can be identified with the probability measure $\mu = \sum_{i=1}^n \frac{1}{n}\delta_{x_i}$. We denote the corresponding identification map by $\iota : X_d \to \mathcal{P}_{fin}(\Omega)$,

$$\iota : (x_1, x_2, \ldots, x_n) \to \sum_{i=1}^n \frac{1}{n}\delta_{x_i}. \tag{4}$$

Then a map $F : X_d \to X_{d'}$ that for any $n$ maps a sequence $(x_1, x_2, \ldots, x_n)$ of $d$-dimensional tokens to a sequence $(y_1, y_2, \ldots, y_n)$ of $d'$-dimensional tokens so that the condition (3) is satisfied, defines a support-preserving map $f : \mathcal{M}_{fin}^+(\Omega) \to \mathcal{M}_{fin}^+(\mathbb{R}^{d'})$ that is the zero-homogeneous extension of the map $f = \iota \circ F \circ \iota^{-1} : \mathcal{P}_{fin}(\Omega) \to \mathcal{P}_{fin}(\mathbb{R}^{d'})$. This map satisfies $f(\sum_{i=1}^n \frac{1}{n}\delta_{x_i}) = \sum_{i=1}^n \frac{1}{n}\delta_{y_i}$. We consider the Wasserstein distance, which is a generalization of the permutation invariant distance of sequences of tokens. We recall that the 1-Wasserstein distance of the measures $\mu = \sum_{i=1}^n \frac{1}{n}\delta_{x_i}$ and $\mu' = \sum_{i=1}^n \frac{1}{n}\delta_{x_i'}$ is given by

$$W_1(\mu, \mu') = \min_\sigma \frac{1}{n} \sum_{i=1}^n |x_i - x'_{\sigma(i)}|,$$

where the minimum is taken over the permutations, $\sigma : \{1, 2, \ldots, n\} \to \{1, 2, \ldots, n\}$. For background material on basic transformers, we refer the reader to Appendix A.4. The convergence of the point measures toward continuous measures as $n \to \infty$, is discussed in Appendix A.3.

**Lemma 2.** *Let $f : \mathcal{M}^+(\Omega) \to \mathcal{M}^+(\mathbb{R}^{d'})$ be a support-preserving map that is continuous in the 1-Wasserstein metric. Then, for any measure of the form (1) with $a_i > 0$ we have that $f(\mu)$ is of the form (2) and satisfies condition (3).*

Lemma 2 can be proved by using sequences of points $x_i$ of which several are equal and simply approximating $a_i$ by rational numbers. The details of the proof Lemma 2 are given in Appendix B.2.

**Definition 2.** *We say that $f : \mathcal{M}^+(\Omega) \to \mathcal{M}^+(\mathbb{R}^{d'})$ is a support-preserving map given by an in-context map, $G : \mathcal{M}^+(\Omega) \times \Omega \to \mathbb{R}^{d'}$, if there exists such a map such that*

$$f(\mu) = G(\mu)_\# \mu,$$

*where $G(\mu)$ is regarded as the map $x \mapsto G(\mu)(x) = G(\mu, x)$. We sometimes write $f = f_G$.*

Note that for a measure $\mu = \sum_{i=1}^n \frac{1}{n}\delta_{x_i}$ it holds that $f_G(\mu) = \sum_{i=1}^n \frac{1}{n}\delta_{y_i}$, where $y_i = G(\mu, x_i)$. A particularly interesting example of such a map is $f_\Gamma : \mu \to \Gamma(\mu)_\# \mu$, where the function $\Gamma$ is a multi-head self attention; see Section 4.1.

In Corollary 1, we revisit the connection between the maps in this definition and transformers. Our goal is to show that a support-preserving map $f$ under a "certain" smoothness condition can be written in the form, $f_G$, with an in-context map, $G$. In the following subsection, we specify this "certain" smoothness in detail.

## 2.2 REGULAR PART OF THE DERIVATIVE

**Definition 3.** *Let $\eta, \rho > 0$. We consider triplets $(\mu, x, \psi) \in \mathcal{X}$, with*

$$\mathcal{X} = \mathcal{X}_{\Omega, \rho, \eta} := \{\mu \in \mathcal{M}^+(\Omega) : \mu(\Omega) \le \rho\} \times \Omega \times \{\psi \in C_0^1(\mathbb{R}^{d'}) : \text{Lip}(\psi) \le \eta\},$$

*which is endowed with the distance function*

$$D_\mathcal{X}\big((\mu_1, x_1, \psi_1), (\mu_2, x_2, \psi_2)\big) = W_1(\mu_1, \mu_2) + |x_1 - x_2| + \|\psi_1 - \psi_2\|_{L^\infty(\mathbb{R}^{d'})}. \tag{5}$$

We observe that $(\mathcal{X}, D_{\mathcal{X}})$ is not a complete metric space (i.e., the Cauchy sequences may not converge in $\mathcal{X}$), as the functions $\psi$ are assumed to be in the space $C_0^1(\mathbb{R}^{d'})$, but we consider their convergence in $L^\infty(\mathbb{R}^{d'})$.

**Definition 4.** *Let $f : \mathcal{M}^+(\Omega) \to \mathcal{M}^+(\mathbb{R}^{d'})$, and $(\mu, x, \psi) \in \mathcal{X}$. We define the $L^\infty$-regular part of the Fréchet derivative of $f$ at $(\mu, x, \psi)$ by the limit*

$$\overline{\mathcal{D}}_f(\mu, x, \psi) := \lim_{k \to \infty} \lim_{\epsilon \to +0} \frac{\langle \psi_k, f(\mu_k + \epsilon \delta_x) - f(\mu_k) \rangle}{\epsilon} \qquad (6)$$

*for all $\psi_k \in C_0^1(\mathbb{R}^{d'})$ and $\mu_k \in \mathcal{M}^+(\Omega)$ such that*

$$\psi_k \text{ is constant in an open neighborhood of } \mathrm{supp}\,(f(\mu_k)) \qquad (7)$$

*and*

$$\lim_{n \to \infty} W_1(\mu_k, \mu) = 0, \quad \lim_{n \to \infty} \|\psi_k - \psi\|_{L^\infty(\mathbb{R}^{d'})} = 0.$$

We note that the existence of the limit $\overline{\mathcal{D}}_f(\mu, x, \psi)$ means that for all $\mu$ and $\psi$, the limits in (6) exist independently of the chosen sequences $\mu_k$ and $\psi_k$.

**2.2.1. Motivational observations.** Let $f_G$ be a support-preserving map given by in-context map $G : \mathcal{M}^+(\Omega) \times \Omega \to \mathbb{R}^{d'}$, where $(\mu, x) \mapsto G(\mu, x)$ is continuous. We observe that for $\mu \in \mathcal{M}^+(\Omega)$, $x \in \Omega$ and $\psi \in C_0^1(\mathbb{R}^{d'})$,

$$\frac{\langle \psi, f_G(\mu + \epsilon \delta_x) - f_G(\mu) \rangle}{\epsilon} = \psi(G(\mu + \epsilon \delta_x, x)) + \int \frac{\psi(G(\mu + \epsilon \delta_x, y)) - \psi(G(\mu, y))}{\epsilon} d\mu(y).$$

Thus the limit as $\epsilon \to +0$ can be written as a sum of two terms

$$\lim_{\epsilon \to +0} \frac{\langle \psi, f_G(\mu + \epsilon \delta_x) - f_G(\mu) \rangle}{\epsilon} = D_{f_G}^{reg}(\mu, x, \psi) + D_{f_G}^{irreg}(\mu, x, \psi),$$

where

$$D_{f_G}^{reg}(\mu, x, \psi) := \lim_{\epsilon \to +0} \psi(G(\mu + \epsilon \delta_x, x)) = \psi(G(\mu, x))$$

and (if the limit exists)

$$D_{f_G}^{irreg}(\mu, x, \psi) := \lim_{\epsilon \to +0} \int \frac{\psi(G(\mu + \epsilon \delta_x, y)) - \psi(G(\mu, y))}{\epsilon} d\mu(y).$$

We call $D_{f_G}^{reg}(\mu, x, \psi)$ the $L^\infty$-regular part of the Fréchet derivative of $f_G$ and $D_{f_G}^{irreg}(\mu, x, \psi)$ the $L^\infty$-irregular part of the Fréchet derivative. This terminology reflects the fact that $\psi \to D_{f_G}^{reg}(\mu, x, \psi)$ is continuous in the $L^\infty$-topology whereas $\psi \to D_{f_G}^{irreg}(\mu, x, \psi)$ is not. The lemma below states that $\overline{\mathcal{D}}_{f_G}(\mu, x, \psi)$ is an extension of the regular part of the derivative $D_{f_G}^{reg}(\mu, x, \psi)$ for functions $G$.

In what follows, we refer to the $L^\infty$-regular part of the Fréchet derivative as the regular part of the derivative. The following lemma is proved in Appendix B.3.

**Lemma 3.** *Let $\Omega \subset \mathbb{R}^d$ be a compact set and let a support-preserving map be given by the in-context map $G : \mathcal{M}^+(\Omega) \times \Omega \to \mathbb{R}^{d'}$, where $(\mu, x) \mapsto G(\mu, x)$ is continuous. Then, for $(\mu, x, \psi) \in \mathcal{X}$,*

$$\overline{\mathcal{D}}_{f_G}(\mu, x, \psi) = D_{f_G}^{reg}(\mu, x, \psi) = \psi(G(\mu, x))$$

*and the map $\mathcal{X} \ni (\mu, x, \psi) \mapsto \overline{\mathcal{D}}_{f_G}(\mu, x, \psi) \in \mathbb{R}$ is uniformly continuous with respect to the metric $D_{\mathcal{X}}$ defined in equation (5).*

As we see in Lemma 3, for map $f_G$ defined with a uniformly continuous in-context function $G$, the regular part of derivative $\overline{\mathcal{D}}_{f_G}(\mu, x, \psi)$ coincides with the above defined object, $D_{f_G}^{reg}(\mu, x, \psi)$ on $\mathcal{X}$. So we consider $D_{f_G}^{reg}(\mu, x, \psi)$ as a new object that is different from the classical Fréchet derivative, and show that the definition of $D_{f_G}^{reg}(\mu, x, \psi)$ can be extended as a generalized regular part of the derivative, $\overline{\mathcal{D}}_f(\mu, x, \psi)$, for a class of functions $f$, for which we do not assume that the classical Fréchet derivative is well-defined. For further remarks on the regular part of derivative $\overline{\mathcal{D}}_f(\mu, x, \psi)$, see Appendix D.

## 3 MAIN RESULT

Our goal is to prove

**Theorem 1.** *Let $\Omega \subset \mathbb{R}^d$ be a compact set. Let $f : \mathcal{M}^+(\Omega) \to \mathcal{M}^+(\mathbb{R}^{d'})$ be a continuous map in the 1-Wasserstein topology. Then,*

   *(A1)* $f$ *is a map given by some in-context map $G$ in the sense of Definition 2, i.e., $f = f_G$ with some function $G : \mathcal{M}^+(\Omega) \times \Omega \to \mathbb{R}^{d'}$; and*

   *(A2) the function $(\mu, x) \to G(\mu, x)$ is continuous,*

*if and only if*

   *(B1)* $f$ *is a support-preserving map in the sense of Definition 1; and*

   *(B2) the regular part of the derivative of $f$, $\overline{\mathcal{D}}_f(\mu, x, \psi)$, exists for all $(\mu, x, \psi) \in \mathcal{X}$, and the map $\mathcal{X} \ni (\mu, x, \psi) \to \overline{\mathcal{D}}_f(\mu, x, \psi) \in \mathbb{R}$ is uniformly continuous with respect to the metric $D_{\mathcal{X}}$ given by Definition 5.*

*Moreover, the map $(\mu, x) \to G(\mu, x)$ is Lipschitz if and only if the map $\mathcal{X} \ni (\mu, x, \psi) \to \overline{\mathcal{D}}_f(\mu, x, \psi) \in \mathbb{R}$ is a Lipschitz map with respect to the metric $D_{\mathcal{X}}$.*

Theorem 1 provides the characterization of support-preserving maps that can be represented by in-context maps through a push forward. Condition (B2) can be roughly described as the uniform continuity of a "certain" derivative of $f$, derived from Definition 4. The continuity of $f$ is not sufficient for the theorem to hold as shown in the following proposition, that is proved in Appendix F.

**Proposition 1.** *Let $d = 1$ and $\Omega = [-3, 3] \subset \mathbb{R}$ and consider the set $\mathcal{P}(\Omega)$ endowed with the 1-Wasserstein topology. There exists a continuous, support-preserving map $f : \mathcal{P}(\Omega) \to \mathcal{P}(\Omega)$ such that there does not exist a continuous map $G : \mathcal{P}(\Omega) \times \Omega \to \Omega$ for which $f = f_G$.*

### 3.1 SKETCH OF THE PROOF OF THEOREM 1: (A1)-(A2) IMPLY (B1)-(B2)

In this section, we give a sketch of the main ideas of proof: Assume that (A1) and (A2) hold true. Then, let $f_G : \mathcal{M}^+(\Omega) \to \mathcal{M}^+(\mathbb{R}^{d'})$ be the map, $f_G(\mu) = G(\mu)_{\#}\mu$, with $G : \mathcal{M}^+(\Omega) \times \Omega \to \mathbb{R}^{d'}$. It is straightforward to prove (B1) and, hence, we will focus on proving that (B2) holds. We assume that $\psi_k, \psi \in C_0^1(\mathbb{R}^{d'})$ and $\mu_k, \mu \in \mathcal{M}_+(\Omega)$, $k = 1, 2, \ldots$ are sequences with $\psi_k$ is constant in an open neighborhood of supp $(f(\mu_k))$ and $\lim_{k \to \infty} W_1(\mu_k, \mu) = 0$, and $\lim_{k \to \infty} \|\psi_k - \psi\|_{L^\infty(\mathbb{R}^{d'})} = 0$. Let

$$\mu_{k,x}^\epsilon := \mu_k + \epsilon \delta_x.$$

Then, by a simple computation,

$$\langle f_G(\mu_{k,x}^\epsilon), \psi \rangle = \int_{\mathbb{R}^d} \psi_k(G(\mu_{k,x}^\epsilon, y)) d\mu_k(y) + \epsilon \psi_k(G(\mu_{k,x}^\epsilon, x)).$$

As the set $\Omega \subset \mathbb{R}^d$ is compact,

$$\mathcal{M}_\rho^+(\Omega) := \{\mu \in \mathcal{M}^+(\Omega) : \mu(\Omega) \leq \rho\},$$

is also compact by the Prokhorov's theorem. Then the map $G : \mathcal{M}_\rho^+(\Omega) \times \Omega \to \mathbb{R}^{d'}$ is uniformly continuous. As $G(\mu_{k,x}^\epsilon, \cdot) \to G(\mu_k, \cdot)$ uniformly in $\Omega \subset \mathbb{R}^d$ as $\epsilon \to 0$, we see that

$$\sup_{y \in \text{supp}(\mu_k)} |G(\mu_{k,x}^\epsilon, y) - G(\mu_k, y)| \to 0 \quad \text{as } \epsilon \to 0.$$

Thus, we find that for sufficiently small $\epsilon \in (0, 1)$

$$\psi_k(G(\mu_{k,x}^\epsilon, y)) = \psi_k(G(\mu_k, y))$$

for all $y \in \text{supp}(\mu_k)$, and

$$\langle f_G(\mu_{k,x}^\epsilon), \psi_k \rangle = \int_{\mathbb{R}^d} \psi_k(G(\mu_k, y)) d\mu_k(y) + \epsilon \psi_k(G(\mu_k, x)).$$

This implies that

$$\overline{\mathcal{D}}_{f_G}(\mu_k, x, \psi_k) = \lim_{\epsilon \to +0} \frac{\langle f_G(\mu_{k,x}^\epsilon), \psi_k \rangle - \langle f_G(\mu_k), \psi_k \rangle}{\epsilon} = \psi_k(G(\mu_k, y)).$$

Upon taking the limit $k \to \infty$, we obtain

$$\overline{\mathcal{D}}_{f_G}(\mu, x, \psi) = \psi(G(\mu, y)).$$

From the uniform (Lipschitz) continuity of $\psi$ and $G$, we can show that the regular part $\overline{\mathcal{D}}_{f_G}$ is uniformly (Lipschitz) continuous with respect to the metric $D_{\mathcal{X}}$. For the details of the proof, see Appendix C.1.

## 3.2 SKETCH OF THE PROOF OF THEOREM 1: (B1)-(B2) IMPLY (A1)-(A2)

Again, here, we give a sketch of the main ideas of the proof. Assume that (B1) and (B2) hold true. Since $f$ is a support-preserving map, $f : \mathcal{M}^+(\Omega) \to \mathcal{M}^+(\mathbb{R}^{d'})$, there are (possibly non-continuous) functions,

$$y_i : \Omega^n \times (0, \infty)^n \to \mathbb{R}^{d'}, \; (\boldsymbol{x}, \boldsymbol{a}) \to y_i(\boldsymbol{x}; \boldsymbol{a}), \quad i = 1, 2, \ldots, n,$$

where $\boldsymbol{x} = (x_1, \ldots, x_n)$ and $\boldsymbol{a} = (a_1, \ldots, a_n)$, such that the following holds: Let $\mu = \sum_{i=1}^n a_i \delta_{x_i} \in \mathcal{M}_{fin}(\Omega)$, $a_i > 0$; then the functions $y_i(\boldsymbol{x}; \boldsymbol{a})$ satisfy

$$f(\mu) = \sum_{i=1}^n a_i \delta_{y_i(\boldsymbol{x};\boldsymbol{a})}.$$

When $\mu \in \mathcal{M}_{fin,dif,(n)}^+(\Omega)$ (which is a refinement of the property that if $j \neq i$ then $a_j \neq a_i$), the functions $(\boldsymbol{x}; \boldsymbol{a}) \to y_i(\boldsymbol{x}; \boldsymbol{a})$ must have the property that if $x_j = x_i$ then $y_j(\boldsymbol{x}; \boldsymbol{a}) = y_i(\boldsymbol{x}; \boldsymbol{a})$.

We have the following lemma, which is proved in Appendix B.4.

**Lemma 4.** *Let $\mu_0 = \sum_{i=1}^n a_i^0 \delta_{x_i^0} \in \mathcal{M}_{fin,dif,(n)}^+(\Omega)$ and $\mu_p = \sum_{i=1}^n a_i^p \delta_{x_i^p} \in \mathcal{M}_{fin,(n)}^+(\Omega)$. Assume that for all $i = 1, 2, \ldots, n$, it holds that $x_i^p \to x_i^0$ and $a_i^p \to a_i^0$ as $p \to \infty$. Then it holds for all $j \in [n]$, that*

$$\lim_{p \to \infty} y_j(\boldsymbol{x}^p; \boldsymbol{a}^p) = y_j(\boldsymbol{x}^0; \boldsymbol{a}^0).$$

We now return to the proof of Theorem 1. Let $\mu \in \mathcal{M}^+(\Omega)$ and $x \in \Omega$, and $\alpha \in C_0^\infty(\mathbb{R}^d)$ be a cutoff function such that $\alpha(x) = 1$ for all $x \in \Omega$ and $\mathrm{Lip}(\alpha(x) \cdot x) \leq \eta$. We define

$$G(\mu, x) := \begin{pmatrix} \overline{\mathcal{D}}_f(\mu, x, \alpha\pi_1) \\ \vdots \\ \overline{\mathcal{D}}_f(\mu, x, \alpha\pi_{d'}) \end{pmatrix},$$

where $\pi_\ell : \mathbb{R}^d \to \mathbb{R}$ is the projection $\pi_\ell(x) = x_\ell$ onto the $\ell$-th component. By (B2), the map $(\mu, x) \mapsto G(\mu, x)$ is continuous, which proves (A2). In what follows, we will prove (A1).

When $\mu \in \mathcal{M}_{fin,dif,(n)}^+(\Omega)$, using Lemma 4, we can prove that for each $j \in [n]$,

$$G(\mu, x_j) = y_j(\boldsymbol{x}; \boldsymbol{a}),$$

which is equivalent to

$$f(\mu) = (G_\mu)_\# \mu \quad \text{for } \mu \in \mathcal{M}_{fin,dif,(n)}^+(\Omega).$$

For the case $\mu \in \mathcal{M}^+(\Omega)$, we choose the sequence $(\tilde{\mu}_k)_{m\in\mathbb{N}} \subset \mathcal{M}_{fin,dif}^+(\Omega)$ such that $\tilde{\mu}_k \to \mu$ as $k \to \infty$, where the limit is considered in the 1-Wasserstein topology (which is possible by Lemma 1). We have already shown that for $\tilde{\mu}_k \in \mathcal{M}_{fin,dif}^+(\Omega)$,

$$f(\tilde{\mu}_k) = (G_{\tilde{\mu}_k})_\# (\tilde{\mu}_k).$$

Hence, by the uniform continuity of $(\mu, x) \mapsto G(\mu, x)$, the limit $k \to \infty$ converges,

$$f(\mu) = \lim_{m \to \infty} (G_{\tilde{\mu}_k})_\# (\tilde{\mu}_k) = (G_\mu)_\# \mu \quad \text{for } \mu \in \mathcal{M}^+(\Omega).$$

For the details of the proof, see Appendix C.2.

## 4 VLASOV FLOWS

Here, we present the close connections between support-preserving maps satisfying (B1) and (B2) in Theorem 1, Vlasov flows and measure-theoretic transformers.

### 4.1 INFINITELY DEEP MEASURE-THEORETIC TRANSFORMERS: UNIVERSAL APPROXIMATION AND THE VLASOV EQUATION

An in-context map as it appears in a single-layer "measure-theoretic" transformer Furuya et al. (2024); Castin et al. (2024) based on multi-head self attention, is of the form,

$$\Gamma : \mathcal{P}(\mathbb{R}^d) \times \mathbb{R}^d \to \mathbb{R}^d, \ \ \Gamma(\mu, x) := x + \sum_{h=1}^{H} W^h \int_{\mathbb{R}^d} \frac{\exp\left(\frac{1}{\sqrt{k}}\langle Q^h x, \ K^h y\rangle\right)}{\int_{\mathbb{R}^d} \exp\left(\frac{1}{\sqrt{k}}\langle Q^h x, \ K^h z\rangle\right) d\mu(z)} V^h y \, d\mu(y),$$

see Appendix A.4 for corresponding functions operating to discrete measures and sequences of tokens. Here, $K^h$ and $Q^h$ are the multi-head key and query matrices in $\mathbb{R}^{k \times d}$, $V^h$ are the multi-head value matrices in $\mathbb{R}^{d_{head} \times d}$, and $W^h$ are the multi-head weight matrices in $\mathbb{R}^{d \times d_{head}}$, respectively. By abuse of notation, $\Gamma(\mu)(x) = \Gamma(\mu, x)$ defines a map $\mathbb{R}^d \to \mathbb{R}^d$. For two in-context maps, $\Gamma_1$ and $\Gamma_2$, the composition $\Gamma_2 \diamond \Gamma_1$ is defined as

$$(\mu, x) \mapsto (\Gamma_2 \diamond \Gamma_1)(\mu, x) := \Gamma_2(\nu, \Gamma_1(\mu, x)), \quad \nu := \Gamma_1(\mu)_\sharp \mu. \tag{8}$$

With this composition, the in-context map, $G_{\text{tran}}$ say, for a multi-layer measure-theoretic transformer is obtained. To be precise, the composition should alternate between in-context maps and context-free MLPs, $F(\mu, x) = F(x)$ say. When restricted to finite discrete empirical measures of the form $\frac{1}{n}\sum_{i=1}^{n} \delta_{x_i}$, $f_{\text{tran}} := f_{G_{\text{tran}}}$ (cf. Definition 2) reduces to a classical transformer acting on a sequence of tokens, $(x_1, \ldots, x_n)$ rather than on a measure $\mu$. For more details, see (Furuya et al., 2024, Section 2). Being based on multi-head self attention, $G_{\text{tran}}$ is (locally) Lipschitz Castin et al. (2024), and, hence, satisfies (A2) in Theorem 1.

As a consequence of Theorem 1, $f_{\text{tran}}$ satisfies (B1) and (B2), while using (Furuya et al., 2024, Theorem 1), we obtain the following universal approximation result that is prove in Appendix B.5.

**Corollary 1.** *Let $f : \mathcal{M}^+(\Omega) \to \mathcal{M}^+(\mathbb{R}^d)$ satisfy (B1) and (B2) in Theorem 1. Then, for any $\epsilon \in (0, 1)$, there exists a sufficiently deep measure-theoretic transformer, $f_{\text{tran}}$, (that is, a deep composition of multi-head self attention maps and MLPs), such that*

$$\sup_{\mu \in \mathcal{P}(\Omega)} W_1(f_{\text{tran}}(\mu), f(\mu)) \le \epsilon.$$

Next, we consider a MLP $F_\eta : \mathbb{R}^d \to \mathbb{R}^d$, see (54) in Appendix A.4, and the attention function

$$\text{Att}_\xi : \mathcal{P}(\mathbb{R}^d) \times \mathbb{R}^d \to \mathbb{R}^d, \quad \text{Att}_\xi(\mu, x) := \sum_{h=1}^{H} W^h \int_{\mathbb{R}^d} \frac{\exp\left(\frac{1}{\sqrt{k}}\langle Q^h x, \ K^h y\rangle\right)}{\int_{\mathbb{R}^d} \exp\left(\frac{1}{\sqrt{k}}\langle Q^h x, \ K^h z\rangle\right) d\mu(z)} V^h y \, d\mu(y).$$

where $\eta$ and $\xi$ are sets of parameter matrices for MLPs and the attention, respectively. Let us write the MLP $F_\eta$ as $F_\eta = Id_x + H_\eta$, and define $\mathcal{V} = \text{Att}_\xi + H_\eta \circ (Id_x + \text{Att}_\xi)$, so that $F_\eta(\Gamma_\xi(\mu, x)) = x + \mathcal{V}(\mu, x)$, see formulas (59) and (60) in Appendix E for detailed formulas. Again, by abuse of notation, $\mathcal{V}(\mu)(x) = \mathcal{V}(\mu, x)$ defines a map or vector field, $\mathbb{R}^d \to \mathbb{R}^d$. We consider layers, $x_i(\tau + 1) = F_{\eta_\tau}(\Gamma_{\xi_\tau}(\mu_i(\tau), x_i(\tau)))$, where the sets $\eta_\tau$ and $\xi_\tau$ of parameter matrices depend on $\tau$. Then, we find that

$$x_i(\tau + 1) - x_i(\tau) = \mathcal{V}_\tau(\mu_\tau)(x_i(\tau)), \quad \text{where } \mu_\tau(.) = \frac{1}{n}\sum_{i=1}^{n} \delta_{x_i(\tau)}(.) \text{ and } \tau = 0, 1, 2, \ldots, T.$$

Taking the continuum limit, scaling $\mathcal{V}_\tau$ with $1/T$, where $T$ signifies the number of layers, and identifying the layer index, $\tau$, with $t \in [0, 1]$ that corresponds to the limit of values $\tau/T$ as $T \to \infty$, the tokens that evolve according to an infinitely deep transformer satisfy

$$\dot{x}_i(t) = \mathcal{V}_t(\mu_t)(x_i(t)) \tag{9}$$

for all $i \in [n]$, where $\mu_t(.) = \frac{1}{n}\sum_{i=1}^{n}\delta_{x_i(t)}(.)$, and $t \in [0,1]$; see also Zhong et al. (2022). This is extended to positive measures by the partial differential, nonlocal transport equation,

$$\partial_t \mu_t + \mathrm{div}(\mathcal{V}_t(\mu_t)\mu_t) \;=\; 0 \quad \text{on } [0,1] \times \mathbb{R}^d, \tag{10}$$

$$\mu_t|_{t=0} \;=\; \mu_0 \quad \text{on } \mathbb{R}^d \tag{11}$$

in the sense of distributions, replacing the (neural) ODE in (9); see Renardy and Rogers (2004). It basically follows from

$$\frac{\mathrm{d}}{\mathrm{d}t}\int_{\mathbb{R}^d}\varphi(t,x)d\mu_t(x) \;=\; \frac{\mathrm{d}}{\mathrm{d}t}\frac{1}{n}\sum_{i=1}^{n}\varphi(t,x_i(t))$$

$$= \int_{\mathbb{R}^d}\left(\partial_t\varphi(t,x) + \langle \nabla_x\varphi(t,x), \mathcal{V}_t(\mu_t)(x)\rangle\right)d\mu_t(x) \tag{12}$$

for all $\varphi \in C_c^\infty([0,1] \times \mathbb{R}^d)$, and integrating by parts. Thus, an infinitely deep measure-theoretic transformer without MLPs, with $\mu_t := f_{\mathrm{tran};t}^\infty(\mu_0)$, $t \in (0,1]$, is argued to satisfy the Vlasov equation; see Piccoli et al. (2015) and Paul and Trélat (2024). Some prior work Sander et al. (2022); Geshkovski et al. (2025); Castin et al. (2025) already discussed that the mean-field (with respect to tokens) and deep transformers are associated with nonlocal transport PDEs. Moreover, the infinitely deep in-context map, $G_{\mathrm{tran};t}^\infty$, satisfies an evolution equation in spacetime that generalizes the equation (9) for the point measures,

$$\partial_t G_{\mathrm{tran};t}^\infty(\mu_0, x) = \mathcal{V}_t(\mu_t)(G_{\mathrm{tran};t}^\infty(\mu_0, x)), \quad G_{\mathrm{tran};0}^\infty(\mu_0, x) = x.$$

### 4.2 THE SOLUTION MAP OF THE VLASOV EQUATION SATISFIES (B1) AND (B2) OF THEOREM 1

Piccoli et al. (2015) studied the well-posedness of nonlocal transport PDEs having the form,

$$\partial_t \mu_t + \mathrm{div}(V(t, \mu_t)\mu_t) = 0, \quad \mu_t|_{t=0} = \mu_0, \tag{13}$$

where $\mu = \mu_t = \mu(t)$ is a time-depending probability measure on $\mathbb{R}^d$ and $V(., \mu) : \mathbb{R} \times \mathbb{R}^d \to \mathbb{R}^d$ is a $C^1-$smooth vector field that depends on $(t, x) \in \mathbb{R} \times \mathbb{R}^d \to \mathbb{R}^d$ and the measure $\mu$. The vector field $V(\mu)$ is called the velocity field.

Under the assumptions on $V$ required by (Piccoli et al., 2015, Theorem 2.3), there exists a unique solution of (13). Moreover, the solution at time $t$, $\mu_t$ can be written as

$$\mu_t = G_t(\mu_0)_{\sharp}\mu_0,$$

where $G_t$ is defined as the unique solution of the following Cauchy problem,

$$\partial_t G_t(\mu_0, x) = V(t, \mu_t)(G_t(\mu_0, x)), \quad G_0(\mu_0, x) = x.$$

Thus, we can define the solution map $f_T : \mathcal{P}(\Omega) \to \mathcal{P}(\mathbb{R}^d)$ (the solution at time $t = T$) by

$$f_T(\mu_0) := \mu_T. \tag{14}$$

The map, $f_T$, is a support-preserving map given by the in-context map, $G_T$. The following proposition is proved Appendix B.6.

**Proposition 2.** *Under the assumptions for $V$ required by (Piccoli et al., 2015, Theorem 2.3), the solution map, $f_T$, defined by (14) satisfies (B1) and (B2). That is, the solution map $f_T$ of the Vlasov flow can be represented as a map $f_{G_T} : \mu \to G_T(\mu)_{\#}\mu$ with a continuous in-context map $G_T$.*

## 5 CONCLUSION AND DISCUSSION

In this work, we fully characterize mappings between measures that can be universally approximated by measure-theoretic transformers. To this end, we introduce a "certain" smoothness condition, which roughly entails that a "certain" derivative of the mapping is uniformly continuous. A limitation of our method is that it is not quantitative. We make rigorous a connection between particle systems and mappings between measures through measure-theoretic transformers in the mean-field regime, which connection has been discussed in various works before. This has implications in the framing of LLMs. Beyond the Vlasov equation, it will be interesting to study the BBGKY hierarchy describing the dynamics of a system of a large number of interacting particles (see, for example, Golse (2016)) with measure-theoretic transformers.

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

# A    NOTATION AND A SUMMARY OF RESULTS OF MEASURE THEORY

## A.1    NOTATIONS

Let $\Omega \subset \mathbb{R}^d$ be a compact set. We denote by $\mathcal{P}(\Omega)$ the space of probability measures on $\Omega$. Below, all measures $\mu$ on subset $\Omega$ of $\mathbb{R}^d$ are defined on the $\sigma$-algebra of the Borel sets of $\Omega$. We denote by $C(\Omega)$ the space of continuous functions from $\Omega$ to $\mathbb{R}$, and the dual coupling between $\varphi \in C(\Omega)$ and $\mu \in \mathcal{P}(\Omega)$ by

$$\langle \varphi, \, \mu \rangle := \int_\Omega \varphi(x) \mathrm{d}\mu(x).$$

With the weak$^*$ topology on $\mathcal{P}(\Omega)$, we have the convergence of sequences of measures,

$$\mu_k \rightharpoonup^* \mu \quad \Leftrightarrow \quad \Big( \forall \varphi \in C_0(\Omega), \, \langle \varphi, \, \mu_k \rangle \to \langle \varphi, \, \mu \rangle \Big).$$

In the case when $\Omega$ is compact, the weak $^*$ topology is equivalent to the topology of the Wasserstein distance $W_p$ $(1 \le p < \infty)$, meaning that

$$\mu_k \rightharpoonup^* \mu \quad \Leftrightarrow \quad W_p(\mu_k, \mu) \to 0,$$

see e.g., (Santambrogio, 2015, Theorem 5.10). By the duality theorem of Kantorovich and Rubinstein, when $\mu, \nu \in \mathcal{P}(\Omega)$, where $\Omega$ is compact, we have that

$$W_1(\mu, \nu) = \sup \left\{ \left. \int_\Omega \varphi(x) \, \mathrm{d}(\mu - \nu)(x) \, \right| \varphi : \Omega \to \mathbb{R} \text{ continuous, } \mathrm{Lip}(\varphi) \le 1 \right\},$$

where

$$\mathrm{Lip}(\varphi) := \sup_{x \ne y} \frac{|\varphi(x) - \varphi(y)|}{|x - y|}$$

denotes the Lipschitz constant for $\varphi : \Omega \to \mathbb{R}$.

We extend $\mathcal{P}(\Omega)$, that is, the set of all probability measures to the set of all strictly positive, finite measures

$$\mathcal{M}^+(\Omega) = \{ s\mu : \, \mu \in \mathcal{P}(\Omega), \, s > 0 \}.$$

We also extend the $W_1$ distance to $\mathcal{M}^+(\Omega)$ by defining for $\mu_1, \mu_2 \in \mathcal{P}(\Omega)$ and $s_1, s_2 > 0$

$$W_1(s_1 \mu_1, s_2 \mu_2) = W_1(\mu_1, \mu_2) + |s_1 - s_2|,$$

see Lombardini and Rossi (2022). Using this extension, we can extend the map $f : \mathcal{P}(\Omega) \to \mathcal{P}(\mathbb{R}^{d'})$ to a map between positive measures invoking $m$-homogeneity $(m \in \mathbb{N}_0)$ according to

$$f(s\mu) = s^m f(\mu) \quad \text{for all } s \in \mathbb{R}_+.$$

We write

$$\mathcal{M}^+_{fin}(\Omega) := \left\{ \sum_{i=1}^n a_i \delta_{x_i} \in \mathcal{M}^+(\Omega) : \, x_i \in \Omega, \, a_i > 0, \, n \in \mathbb{N} \right\},$$

and

$$\mathcal{M}^+_{fin,(n)}(\Omega) := \left\{ \sum_{i=1}^n a_i \delta_{x_i} \in \mathcal{M}^+(\Omega) : \, x_i \in X, \, a_i > 0 \right\}.$$

Finally, we denote by $\mathcal{M}^+_{fin,dif,(n)}(\Omega)$ the measures of the form

$$\mu = \sum_{i=1}^n a_i \delta_{x_i} \in \mathcal{M}_{fin,(n)}(\Omega),$$

where $a_j > 0$ and for all non-empty subsets $J, K \subset \{1, 2, \ldots, n\}$ satisfying $J \cap K = \emptyset$ it holds that

$$\sum_{j \in J} a_j \ne \sum_{k \in K} a_k.$$

We set $\mathcal{M}^+_{fin,dif}(\Omega) = \bigcup_{n=1}^\infty \mathcal{M}^+_{fin,dif,(n)}(\Omega)$. For $\mu \in \mathcal{M}^+_{fin,dif,(n)}(\Omega)$ we define the minimal gap

$$\mathrm{gap}(\mu) = \min_{J,K \subset \{1,2,\ldots,n\}, J \cap K = \emptyset, \, J \ne \emptyset} \left| \sum_{j \in J} a_j - \sum_{k \in K} a_k \right|. \tag{15}$$

## A.2 PUSH FORWARDS OF MEASURES

We will consider push forwards of measures in various maps. When $\nu$ is a general Borel measure on set $\Omega \subset \mathbb{R}^d$ and $F : \Omega \to \mathbb{R}^{d'}$ is a continuous map, the push-forward measure of $\nu$ in the map $F$, denoted by $F_{\#}\nu$, is the measure that for an open (or Borel measurable) set $A$ is defined to be

$$F_{\#}\nu(A) = \nu(F^{-1}(A)).$$

When $\mu = \sum_{j=1}^n a_j \delta_{x_j}$ is a discrete measure supported at points $x_1, \ldots, x_n$, we have

$$F_{\#}\mu = \sum_{j=1}^n a_j \delta_{y_j}, \quad y_j = F(x_j).$$

When $\nu = \rho(x)dx$ is a continuous measure where $\rho : \mathbb{R}^d \to [0, \infty)$ is a continuous function and $dx$ is the Lebesgue measure on $\mathbb{R}^d$ and $F : \mathbb{R}^d \to \mathbb{R}^d$ is a differentiable map which inverse function $F^{-1} : \mathbb{R}^d \to \mathbb{R}^d$ is differentiable, then

$$F_{\#}(\rho(x)dx) = \tau(x)dx, \quad \text{where } \tau(x) = \rho(F^{-1}(x)) \cdot \left| \det\left( \frac{\partial F}{\partial x}(F^{-1}(x)) \right) \right|,$$

where $\det(\frac{\partial F}{\partial x}(F^{-1}(x)))$ is the determinant of the Jacobian matrix of the function $F$ evaluated at the point $F^{-1}(x)$.

When $F : \mathbb{R}^d \to \mathbb{R}^{d'}$ is a smooth injective map and $d' > d$, the push forward of the measures $\mu$ on $\mathbb{R}^d$ to the $d$-dimensional image manifold $M = F(\mathbb{R}^d)$ of $F$ are discussed e.g. in Kothari et al. (2021).

## A.3 CONVERGENCE OF POINT MEASURES TO A GENERAL MEASURE

Let us consider the convergence of discrete measures $\mu_n = \sum_{j=1}^n a_{n,j} \delta_{x_{n,j}}$ to continuous measures. Let $\Omega \subset \mathbb{R}^d$ be a compact set, $x_{n,j} \in \Omega$, and $a_{n,j} > 0$ are such that $\sum_{j=1}^n a_{n,j} = 1$. If for all relatively open subsets $U \subset \Omega$ there exists limits

$$m(U) = \lim_{n \to \infty} \mu_n(U), \quad \text{where } \mu_n(U) = \sum_{x_{n,j} \in U} a_{n,j}, \tag{16}$$

then the limits $m(U)$ define a (Borel) probability measure in $m \in \mathcal{P}(\Omega)$ and the measures $\mu_n$ converge in the 1-Wasserstein topology to the measure $m$.

By the Portmanteau theorem, see Klenke (2020), Theorem 13.16 (see also Remark 13.14), the existence of limits (16) is equivalent to following conditions:

(C1) There is a probability measure $m \in \mathcal{P}(\Omega)$ such that $m(U) \geq \liminf_{n \to \infty} \mu_n(U)$ for all relatively open sets $U \subset \Omega$

(C2) There is a probability measure $m \in \mathcal{P}(\Omega)$ such that for all Lipschitz functions $\phi : \Omega \to \mathbb{R}$

$$\int_{\Omega} \phi \, d\mu_n = \sum_{j=1}^n a_{n,j} \phi(x_{n,j}) \to \int_{\Omega} \phi \, dm, \quad \text{as } n \to \infty, \tag{17}$$

that is, the existence of limits $m(U)$ in (16) and the conditions (C1) and (C2) are all equivalent to that $\mu_n$ converge weakly to $m$ that is further equivalent to that $\mu_n$ converge to $m$ in the 1-Wasserstein topology.

In particular, consider the case when $x_{n,j} = x_j$ are independent of $n$ and $a_{n,j} = 1/n$. Also, let us consider the Lipschitz functions $\phi : \Omega \to \mathbb{R}$ as feature functions. That is the measures, $\mu_n$ are the point measures

$$\mu_n = \sum_{j=1}^n \frac{1}{n} \delta_{x_j}$$

that correspond to prompts $X_n = (x_1, x_2, \ldots, x_n)$, that is, sequences of $n$ tokens. Then, if the the prompt length $n$ goes to infinity, if follows from Prokhorov's theorem (Klenke, 2020, Theorem

13.29 and Corollary 13.30), that there is at least one sub-sequence $X_{n_k}$ of prompts, where $n_k \to \infty$ as $k \to \infty$ such that for all feature functions $\phi \in C^{0,1}(\Omega)$ the averages of the features

$$\int_\Omega \phi d\mu_{n_k} = \sum_{j=1}^{n_k} \frac{1}{n} \phi(x_j)$$

converge to some limit

$$\lim_{k\to\infty} \int_\Omega \phi d\mu_{n_k} = \int_\Omega \phi d\mu,$$

These limits define a probability measure $\mu \in \mathcal{P}(\Omega)$ such that

$$\lim_{k\to\infty} W_1(\mu_{n_k}, \mu) = 0. \tag{18}$$

Moreover, by (Reed and Simon, 1980, Theorems I.13 and I.14), the measure $\mu$ can be written as a sum of three measures,

$$\mu = \nu_1 + \nu_2 + \nu_3, \quad \nu_1 = \sum_{i=1}^N a_j \delta_{y_j}, \quad \nu_2 = \rho(x)d\mathbf{x}, \quad \nu_3 \perp d\mathbf{x} \tag{19}$$

where $\nu_1$ is a pure point measure supported at the points $y_j \in \Omega$ with $N \in \mathbb{N} \cup \{\infty\}$ and $a_j > 0$, $\nu_2$ is an absolutely continuous measure having the density $\rho(x)$ with respect to the Lebesgue measure $d\mathbf{x}$ of $\mathbb{R}^d$, and $\nu_3$ is a singular continuous measure, that is, there is a set $S \subset \Omega$ which the Lebesgue measure is zero such that $\nu_3(\Omega \setminus S) = 0$ and $\nu_3(\{p\}) = 0$ for all singleton sets with $p \in \Omega$.

### A.4 ATTENTION AND TRANSFORMERS

Finally we recall notations related to attention functions. The multi-head self attention is the function

$$\Gamma : \mathcal{P}(\mathbb{R}^d) \times \mathbb{R}^d \to \mathbb{R}^d,$$

$$\Gamma(\mu, x) = x + \sum_{h=1}^H W^h \int_{\mathbb{R}^d} \frac{\exp\left(\frac{1}{\sqrt{k}}\langle Q^h x, K^h y\rangle\right)}{\int_{\mathbb{R}^d} \exp\left(\frac{1}{\sqrt{k}}\langle Q^h x, K^h z\rangle\right) d\mu(z)} V^h y \, d\mu(y) \tag{20}$$

$$= x + Att(\mu, x).$$

We recall that here $K^h$ and $Q^h$ are the multi-head key and query matrices in $\mathbb{R}^{k\times d}$, $V^h$ are the multi-head value matrices in $\mathbb{R}^{d_{head}\times d}$, and $W^h$ are the multi-head weight matrices in $\mathbb{R}^{d\times d_{head}}$, respectively. When

$$\mu = \sum_{i=1}^n \frac{1}{n} \delta_{x_i} \tag{21}$$

is a discrete measure corresponding to a sequence $x_1, x_2, \ldots, x_n$ of points in $\Omega \subset \mathbb{R}^d$, it holds that

$$\Gamma(\mu, x) = x + \sum_{h=1}^H W^h \int_{\mathbb{R}^d} \frac{\exp\left(\frac{1}{\sqrt{k}}\langle Q^h x, K^h y\rangle\right)}{\int_{\mathbb{R}^d} \exp\left(\frac{1}{\sqrt{k}}\langle Q^h x, K^h z\rangle\right) d\mu(z)} V^h y \, d\mu(y)$$

$$= x + \sum_{h=1}^H W^h \sum_{\ell=1}^n \frac{\exp\left(\frac{1}{\sqrt{k}}(Q^h x)^\top (K^h x_\ell)\right)}{\sum_{j=1}^n \exp\left(\frac{1}{\sqrt{k}}(Q^h x)^\top (K^h x_j)\right)} V^h x_\ell, \tag{22}$$

where $v^\top$ denotes the transpose of a column vector $v \in \mathbb{R}^k$.

For the measure $\mu$ given in (21) it holds that

$$f_\Gamma\left(\sum_{i=1}^n \frac{1}{n} \delta_{x_i}\right) = \Gamma(\mu, \cdot)_\# \mu = \sum_{i=1}^n \frac{1}{n} \delta_{y_i}, \tag{23}$$

where $y_i = \Gamma(\mu, x_i)$.

In the case that the measures $\mu_{n_k} = \sum_{i=1}^{n_k} \frac{1}{n} \delta_{x_i}$ converge in 1-Wasserstein topology to a measure $\mu$ as $k \to \infty$, we have pointwise limits

$$\lim_{k \to \infty} \Gamma(\mu_{n_k}, x) = \Gamma(\mu, x), \tag{24}$$

where $\Gamma(\mu_{n_k}, x)$ and $\Gamma(\mu, x)$ are given in formulas (22) and (20), respectively. Moreover, it holds that the push forwards of the measures satisfy the limit

$$\lim_{k \to \infty} \Gamma(\mu_{n_k}, \cdot)_{\#} \mu_{n_k} = \Gamma(\mu, \cdot)_{\#} \mu \tag{25}$$

in the 1-Wasserstein topology.

Let us next consider the prompts $(x_1, x_2, \ldots, x_n)$ and the corresponding discrete measures $\mu_n = \sum_{i=1}^{n} \frac{1}{n} \delta_{x_i}$. As seen above, then there exists at least one sub-sequence $\mu_{n_k}$ that converge to a general probability measure $\mu \in \mathcal{P}(\Omega)$, that is a sum of a point measure, a continuous measure, and a measure that are singular with respect to the standard measure of $\mathbb{R}^d$, see formula (19). Thus, to understand properties of transformers it is useful to consider mappings between general probability measures that have the same properties of the transformers.

## B PROOFS FOR TECHNICAL PARTS

### B.1 PROOF OF LEMMA 1

*Proof.* Let $\mu \in \mathcal{M}^+(\Omega)$ and let $\epsilon \in (0, 1)$. Since $\mathcal{M}^+_{fin}(\Omega)$ is dense in $\mathcal{M}^+(\Omega)$ in 1-Wasserstein topology, there is $\mu_k \in \mathcal{M}^+_{fin,dif}(\Omega)$ with $\mu_k = \sum_{i=1}^{n} a_i \delta_{x_i}, a_i > 0$ such that

$$W_1(\mu_k, \mu) \le \epsilon.$$

We can choose $\tilde{a}_1, \ldots, \tilde{a}_n > 0$ such that $|\tilde{a}_j - a_j| < \epsilon/n$ and, for any non-empty disjoint subsets $J, K \subset \{1, \ldots, n\}$, it holds that

$$\sum_{i \in J} \tilde{a}_i \ne \sum_{i \in K} \tilde{a}_i.$$

Indeed, setting $\tilde{a}_i = a_i + \eta_i$, the equality

$$\sum_{i \in J} \tilde{a}_i = \sum_{i \in K} \tilde{a}_i,$$

is equivalent to

$$\sum_{i \in J} \eta_i - \sum_{i \in K} \eta_i = \underbrace{\sum_{i \in K} a_i - \sum_{i \in J} a_i}_{=: \Delta_{J,K}}.$$

Since the set $\cup_{J,K} \{\eta \in \mathbb{R}^n : \sum_{i \in J} \eta_i - \sum_{i \in K} \eta_i = \Delta_{J,K}\}$ of affine hyperplanes are measure-zero set, we can choose small $|\eta_i| < \epsilon/n$ so that

$$\eta \notin \bigcup_{J,K} \left\{ \eta \in \mathbb{R}^n : \sum_{i \in J} \eta_i - \sum_{i \in K} \eta_i = \Delta_{J,K} \right\}.$$

Thus, defining by $\tilde{\mu}_n = \sum_{i=1}^{n} \tilde{a}_i \delta_{x_i} \in \mathcal{M}^+_{fin,dif}(\Omega)$, we see that

$$W_1(\mu_k, \tilde{\mu}_n) < \epsilon.$$

We have proved Lemma 1. $\qquad\square$

### B.2 PROOF OF LEMMA 2

*Proof.* When $\tilde{x}_j \in \Omega, \tilde{a}_j > 0, j = 1, 2, \ldots, \tilde{n}$ are of the form $\tilde{a}_j = cm_j$ where $c > 0$ and $m_j \in \mathbb{Z}_+$, we can can write the measure

$$\mu = \sum_{j=1}^{\tilde{n}} \tilde{a}_j \delta_{\tilde{x}_j} \tag{26}$$

in the form

$$\mu = \sum_{i=1}^{n} \frac{\mu(\Omega)}{n} \delta_{x_i}, \tag{27}$$

where $n = \sum_{j=1}^{\tilde{n}} m_k$ and $x_1, x_2 \ldots, x_n$ is a sequence where each point $\tilde{x}_j$ appears $m_j$ times. As $f$ is a support preserving map, there are $y_i \in \mathbb{R}^{d'}$, $i = 1, 2, \ldots, n$, such that

$$f(\mu) = \sum_{i=1}^{n} \frac{\mu(\Omega)}{n} \delta_{y_i}. \tag{28}$$

Moreover, $y_{i_1} = y_{i_2}$ if $x_{i_1} = x_{i_2}$. Hence, we can write $f(\mu)$ in the form

$$f(\mu) = \sum_{j=1}^{\tilde{n}} \left( \sum_{i:\, x_i = \tilde{x}_j} \frac{\mu(\Omega)}{n} \right) \delta_{y_i}$$

$$= \sum_{j=1}^{\tilde{n}} \frac{cm_j}{n} \delta_{\tilde{y}_j} \tag{29}$$

where $c = \mu(\Omega)$ and the set $\{\tilde{y}_1, \ldots, \tilde{y}_{\tilde{n}}\}$ contains the same points as the set $\{y_1, \ldots, y_n\}$. Below, we denote $\tilde{X} = (\tilde{x}_1, \ldots, \tilde{x}_{\tilde{n}})$ and $Y_j(\tilde{X}, \mu) := \tilde{y}_j$. The above shows that the claim is valid when the $a_j$ are of the form $a_j = cm_j$, $m_j \in \mathbb{Z}_+$.

We now consider general values, $a_i > 0$, and points, $x_i \in \Omega$, $i = 1, \ldots, n$ and let $c = \sum_{i=1}^{n} a_i$. We let $N_k, m_{k,i} \in \mathbb{Z}_+$, $i = 1, \ldots, n$, $k \in \mathbb{Z}_+$ be such that

$$\lim_{k \to \infty} \frac{m_{k,i}}{N_k} = a_i \quad \text{for all } i = 1, 2, \ldots, n.$$

Also, we let $c_k = c/N_k$ and

$$\mu_k = \sum_{i=1}^{n} c_k m_{k,i} \delta_{x_i}. \tag{30}$$

We write $X = (x_1, \ldots, x_n)$. Then, as we have already shown that the claim is valid for measures $\mu_k$ of the form (30), we can write $f(\mu_k)$ as

$$f(\mu_k) = \sum_{i=1}^{n} c_k m_{k,i} \delta_{Y_i(X, \mu_k)} = \sum_{i=1}^{n} \frac{m_{k,i}}{N_k} \delta_{Y_i(X, \mu_k)} \in \mathcal{M}_{fin,(n)}^{+}(\Omega). \tag{31}$$

As $f$ is a continuous map in the 1-Wasserstein topology and the set $\mathcal{M}_{fin,(n)}^{+}(\Omega)$ is a closed subset of $\mathcal{M}^{+}(\Omega)$ in the same topology and $\mathcal{M}^{+}(\Omega)$ is a complete space, we conclude that there exists a limit

$$f(\mu) = \lim_{k \to \infty} f(\mu_k) \in \mathcal{M}_{fin,(n)}^{+}(\Omega). \tag{32}$$

Thus we can write $f(\mu)$ in the form,

$$f(\mu) = \sum_{j=1}^{n'} b_j \delta_{z_j}, \tag{33}$$

with some $n' \leq n$, $z_j \in \Omega$ and $b_j > 0$. We choose

$$\rho = \min\{|z_{j=1} - z_{j_2}| :\, j_1, j_2 \in [n'],\ j_1 \neq j_2\} > 0$$

and let $A = \min_j a_j > 0$. Moreover, as $f(\mu_k) \to f(\mu)$ in the 1-Wasserstein metric as $k \to \infty$, we observe that for each $k$ there is a partition of the set $\{1, 2, \ldots, n\}$ to a union of disjoint sets, $I_{1,k}, \ldots, I_{n',k}$, such that when $k$ is sufficiently large,

$$\sum_{j=1}^{n'} \frac{A}{4} \min_{i \in I_{j,k}} \text{dist}(Y_i(X, \mu_k), z_j) + \frac{1}{4} \sum_{j=1}^{n'} \left| \left( \sum_{i \in I_{j,k}} \frac{m_{k,i}}{N_k} \right) - b_j \right| \leq W_1(f(\mu_k), f(\mu)).$$

As $W_1(f(\mu_k), f(\mu)) \to 0$ as $k \to \infty$, we find that by replacing $\mu_k$ by its suitable subsequence, we can assume that the partition $I_{1,k}, \ldots, I_{n',k}$ is equal to a partition $I_1, \ldots, I_{n'}$ that is independent of $k$, and

$$Y_i(X, \mu_k) \to z_i, \quad \text{as } k \to \infty. \tag{34}$$

Moreover,

$$\sum_{i \in I_j} a_i = \sum_{i \in I_j} \lim_{k \to \infty} \frac{m_{k,i}}{N_k} = b_j \tag{35}$$

for $j = 1, 2, \ldots, n'$. Then $b_j = \sum_{i \in I_j} a_i$, and

$$f(\mu) = \sum_{j=1}^{n'} \left( \sum_{i \in I_j} a_i \right) \delta_{z_j} = \sum_{i=1}^{n} a_i \delta_{y_i}, \tag{36}$$

where $y_1, \ldots, y_n$ is a sequence of the points $z_1, \ldots, z_{n'}$ where each $z_j$ appears $|I_j|$ times. This proves the claim for general weights $a_i > 0$. □

### B.3 Proof of Lemma 3

*Proof.* Let $f_G : \mathcal{M}^+(\Omega) \to \mathcal{M}^+(\mathbb{R}^{d'})$ be the map, $f_G(\mu) = G(\mu)_{\#}\mu$, with a continuous map $G : \mathcal{M}^+(\Omega) \times \Omega \to \mathbb{R}^{d'}$. We assume that $\psi_k, \psi \in C_0^1(\mathbb{R}^{d'})$ and $\mu_k, \mu \in \mathcal{M}_+(\Omega)$, $k = 1, 2, \ldots$ are sequences with

$$\psi_k \text{ is constant in an open neighborhood of supp}\,(f(\mu_k))$$

and

$$\lim_{k \to \infty} W_1(\mu_k, \mu) = 0, \quad \lim_{k \to \infty} \|\psi_k - \psi\|_{L^\infty(\mathbb{R}^{d'})} = 0.$$

Let

$$\mu_{k,x}^\epsilon := \mu_k + \epsilon \delta_x.$$

Then, by the simple computation,

$$\langle f_G(\mu_{k,x}^\epsilon), \psi \rangle = \int_{\mathbb{R}^d} \psi_k(G(\mu_{k,x}^\epsilon, y)) d\mu_k(y) + \epsilon \psi_k(G(\mu_{k,x}^\epsilon, x)).$$

As the set $\Omega \subset \mathbb{R}^d$ is compact,

$$\mathcal{M}_\rho^+(\Omega) := \{\mu \in \mathcal{M}^+(\Omega) : \mu(\Omega) \le \rho\}$$

is also compact by the Prokhorov's theorem. Then, the map $G : \mathcal{M}_\rho^+(\Omega) \times \Omega \to \mathbb{R}^{d'}$ is uniformly continuous. As $G(\mu_{k,x}^\epsilon, \cdot) \to G(\mu_k, \cdot)$ uniformly in $\Omega \subset \mathbb{R}^d$ as $\epsilon \to 0$, we see that

$$\sup_{y \in \text{supp}(\mu_k)} |G(\mu_{k,x}^\epsilon, y) - G(\mu_k, y)| \to 0 \quad \text{as } \epsilon \to 0.$$

Thus, we find that for sufficiently small $\epsilon \in (0, 1)$

$$\psi_k(G(\mu_{k,x}^\epsilon, y)) = \psi_k(G(\mu_k, y))$$

for all $y \in \text{supp}\,(\mu_k)$, and

$$\langle f_G(\mu_{k,x}^\epsilon), \psi_k \rangle = \int_{\mathbb{R}^d} \psi_k(G(\mu_k, y)) d\mu_k(y) + \epsilon \psi_k(G(\mu_k, x)).$$

This implies that

$$\overline{\mathcal{D}}_{f_G}(\mu_k, x, \psi_k) = \lim_{\epsilon \to +0} \frac{\langle f_G(\mu_{k,x}^\epsilon), \psi_k \rangle - \langle f_G(\mu_k), \psi_k \rangle}{\epsilon} = \psi_k(G(\mu_k, y)).$$

Upon taking the limit $k \to \infty$, we obtain

$$\overline{\mathcal{D}}_{f_G}(\mu, x, \psi) = \psi(G(\mu, y)).$$

Next, we prove that the map $(\mu, y, \psi) \to \overline{\mathcal{D}}_{f_G}(\mu, y, \psi)$ is uniformly continuous. Let $\epsilon_1 > 0$. By the uniform continuity of $G$, there is a $\delta_1 = \delta_1(\epsilon_1) \in (0, \epsilon_1)$ such that if $W_1(\mu_1, \mu_2) < \delta_1(\epsilon_1)$ and $|y_1 - y_2| < \delta_1(\epsilon_1)$ then $|G(\mu_1, y_1) - G(\mu_2, y_2)| < \epsilon_1/2$. Let $(\mu_1, y_1, \psi_1), (\mu_2, y_2, \psi_2) \in \mathcal{X}$ so that $\mathrm{Lip}(\psi_j) \leq \eta$ for $j = 1, 2$. Also, assume that $\|\psi_1 - \psi_2\|_{L^\infty} < \delta_1(\epsilon_1)$. We then see that

$$
\begin{aligned}
|\overline{\mathcal{D}}_{f_G}(\mu_1, y_1, \psi_1) - \overline{\mathcal{D}}_{f_G}(\mu_2, y_2, \psi_2)| &= |\psi_1(G(\mu_1, y_1)) - \psi_2(G(\mu_2, y_2))| \\
&\leq |\psi_1(G(\mu_1, y_1)) - \psi_1(G(\mu_2, y_2))| \\
&\quad + |\psi_1(G(\mu_2, y_2)) - \psi_2(G(\mu_2, y_2))| \\
&\leq \mathrm{Lip}(\psi_1)|G(\mu_1, y_1) - G(\mu_2, y_2)| + \|\psi_1 - \psi_2\|_{L^\infty} \\
&\leq \mathrm{Lip}(\psi_1)\epsilon_1 + \delta_1(\epsilon_1) \\
&\leq (\eta + 1)\epsilon_1.
\end{aligned}
$$

We observe that if $D_{\mathcal{X}}((\mu_1, y_1, \psi_1), (\mu_2, y_2, \psi_2)) < \delta_1(\epsilon_1)$ then $W_1(\mu_1, \mu_2) < \delta_1(\epsilon_1)$ and $|y_1 - y_2| < \delta_1(\epsilon_1)$, and moreover that $\|\psi_1 - \psi_2\|_{L^\infty} < \delta_1(\epsilon_1)$. We conclude that $(\mu, y, \psi) \to \overline{\mathcal{D}}_{f_G}(\mu, y, \psi)$ is uniformly continuous. $\qquad\square$

### B.4 PROOF OF LEMMA 4

*Proof.* The assumptions imply that $W_1(\mu_p, \mu_0) \to 0$ as $p \to \infty$. Hence, as $f$ is continuous in the 1-Wasserstein distance, it holds that $W_1(f(\mu_p), f(\mu_0)) \to 0$ as $p \to \infty$. If the claim is not valid, there are $k$ and $(\boldsymbol{x}^p, \boldsymbol{a}^p)$ such that $(\boldsymbol{x}^p, \boldsymbol{a}^p) \to (\boldsymbol{x}^0, \boldsymbol{a}^0)$ as $p \to \infty$ and $\mu_0 = \sum_{i=1}^n a_i^0 \delta_{x_i^0} \in \mathcal{M}_{fin,dif,(n)}(\Omega)$, and the sequence $y_k(\boldsymbol{x}^p; \boldsymbol{a}^p)$, $p \in \mathbb{Z}_+$, does not converge to the value $y_k(\boldsymbol{x}^0; \boldsymbol{a}^0)$ as $p \to \infty$. By replacing $(\boldsymbol{x}^p; \boldsymbol{a}^p)$ by its suitable subsequence, we can assume that there exists $z \in \mathbb{R}^{d'}$ such that

$$
\lim_{p \to \infty} y_k(\boldsymbol{x}^p; \boldsymbol{a}^p) = z \neq y_k(\boldsymbol{x}^0; \boldsymbol{a}^0). \tag{37}
$$

As all $a_i^0$ are strictly positive and $a_i^p \to a_i^0$, there are $b > 0$ and $p_0$ such that we have $a_i^p > b$ for all $p > p_0$ and $i$. As $f(\mu_p) = \sum_{k=1}^n a_k^p \delta_{y_k(\boldsymbol{x}^p; \boldsymbol{a}^p)} \to f(\mu_0)$ in 1-Wasserstein distance, we see that

$$
\lim_{p \to \infty} \sup_{y \in \mathrm{supp}(f(\mu_p))} \mathrm{dist}(y, \mathrm{supp}\,(f(\mu_0))) = 0.
$$

This and (37) imply that there is $k_0 \neq k$ such that

$$
\lim_{p \to \infty} y_k(\boldsymbol{x}^p; \boldsymbol{a}^p) = y_{k_0}(\boldsymbol{x}^0; \boldsymbol{a}^0) \neq y_k(\boldsymbol{x}^0; \boldsymbol{a}^0). \tag{38}
$$

Then, as (38) holds, we find that

$$
\lim_{p \to \infty} |y_k(\boldsymbol{x}^p; \boldsymbol{a}^p) - y_k(\boldsymbol{x}^0; \boldsymbol{a}^0)| \geq \min\{|y - y'| : \ y, y' \in \mathrm{supp}\,(f(\mu_0)), \ y \neq y'\} > 0
$$

and that the measures $\mu_p = \sum_{i=1}^n a_i^p \delta_{x_i^p}$ and $\mu_0 = \sum_{i=1}^n a_i^0 \delta_{x_i^0}$ and their images under $f$, that is,

$$
f(\mu_p) = \sum_{i=1}^n a_i^p \delta_{y_i(\boldsymbol{x}^p; \boldsymbol{a}^p)} \quad \text{and} \quad f(\mu_0) = \sum_{i=1}^n a_i^0 \delta_{y_i(\boldsymbol{x}^0; \boldsymbol{a}^0)},
$$

satisfy the inequality

$$
\lim_{p \to \infty} W_1(f(\mu_p), f(\mu_0)) \geq \mathrm{gap}(\mu_0) \min\{|y - y'| : \ y, y' \in \mathrm{supp}\,(f(\mu_0)), \ y \neq y'\} > 0,
$$

where $\mathrm{gap}(\mu_0)$ is defined in (15). This is not possible in view of the 1-Wasserstein continuity of $f$. Thus, the claim follows. $\qquad\square$

### B.5 PROOF OF COROLLARY 1

*Proof.* Using Theorem 1, there is an in-context map $G$ such that $f(\mu) = G(\mu)_\sharp \mu$. Since the map $G$ is continuous, by using (Furuya et al., 2024, Theorem 1), for any $\epsilon \in (0, 1)$, there is a measure-theoretic transformer-style in-context mapping $G_{\mathrm{tran}} := F_{\xi_L} \diamond \Gamma_{\theta_L} \diamond \ldots \diamond F_{\xi_1} \diamond \Gamma_{\theta_1}$ such that

$$
\sup_{(\mu, x) \in \mathcal{P}(\Omega) \times \Omega} |G_{\mathrm{tran}}(\mu, x) - G(\mu, x)| \leq \epsilon,
$$

which implies that, by the duality theorem of Kantorovich and Rubinstein,

$$W_1(f_{\text{tran}}(\mu), f(\mu)) = \sup_{\text{Lip}(\varphi) \leq 1} \int \varphi(G_{\text{tran}}(\mu, x)) - \varphi(G(\mu, x)) d\mu(x)$$

$$\leq \int |G_{\text{tran}}(\mu, x) - G(\mu, x)| d\mu(x) \leq \epsilon.$$

We have proved Corollary 1. $\qquad\square$

### B.6 PROOF OF PROPOSITION 2

*Proof.* By (Piccoli et al., 2015, Theorem 2.3), there exists $G_t : \mathcal{P}(\Omega) \times \Omega \to \mathbb{R}^d$ such that

$$\mu_t = G_t(\mu_0)_\sharp \mu_0,$$

where $G_t$ is defined by the unique solution of the following Cauchy problem

$$\partial_t G_t(\mu_0, x) = V[\mu(t)](t, G_t(\mu_0, x)), \quad G_0(\mu_0, x) = x. \tag{39}$$

This is a push forward, thus the solution map, $f$, satisfies (B1). Moreover, if the map $(\mu, x) \mapsto G_T(\mu, x)$ is Lipschitz continuous, by Lemma 3 the map $(\mu, x, \psi) \mapsto \overline{\mathcal{D}}_{f_T}(\mu, x, \psi)$ is Lipschitz continuous with respect to the metric $D_\mathcal{X}$. This implies (B2). In what follows, we will prove that the map $(\mu, x) \mapsto G_T(\mu, x)$ is Lipschitz continuous.

We estimate, for $\mu_0, \nu_0 \in \mathcal{P}(\Omega)$ and $x, y \in \Omega$,

$$\frac{d}{dt} \|G_t(\mu_0, x) - G_t(\nu_0, y)\|_2$$

$$\leq \left\| \frac{d}{dt} G_t(\mu_0, x) - \frac{d}{dt} G_t(\nu_0, y) \right\|_2 = \|V[\mu(t)](t, G_t(\mu_0, x)) - V[\nu(t)](t, G_t(\nu_0, x))\|_2$$

$$\leq \|V[\mu(t)](t, G_t(\mu_0, x)) - V[\mu(t)](t, G_t(\nu_0, x))\|_2$$

$$\qquad + \|V[\mu(t)](t, G_t(\nu_0, x)) - V[\nu(t)](t, G_t(\nu_0, x))\|_2$$

$$\leq L(t) \|G_t(\mu_0, x) - G_t(\nu_0, x)\|_2 + K(t) W_1(\mu(t), \nu(t))$$

$$\leq L(t) \|G_t(\mu_0, x) - G_t(\nu_0, x)\|_2 + K(t) e^{C_T t} W_1(\mu_0, \nu_0),$$

for some $K, L \in L^\infty_{loc}(\mathbb{R})$ and some $C_T > 0$. Here, we have used the assumption of Lipschitz continuity required in (Piccoli et al., 2015, Theorem 2.3), and the stability estimate (Piccoli et al., 2015, (2.3)). By Gronwall's inequality, we find that

$$\|G_t(\mu_0, x) - G_t(\nu_0, x)\|_2$$

$$\leq e^{A(t)} \underbrace{\|G_0(\mu_0, x) - G_0(\nu_0, x)\|_2}_{= \|x-y\|_2} + \left( \int_0^t e^{A(t)-A(s)} K(s) e^{C_T s} ds \right) W_1(\mu_0, \nu_0),$$

where $A(t) = \int_0^t L(s) ds$. Thus, substituting $t = T$, there exists $C_T' > 0$ such that

$$\|G_T(\mu_0, x) - G_T(\nu_0, x)\|_2 \leq C_T'(W_1(\mu_0, \nu_0) + \|x - y\|_2),$$

which implies that the map $(\mu, x) \mapsto G_T(\mu, x)$ is Lipschitz continuous. $\qquad\square$

## C PROOF OF THEOREM 1

### C.1 PART 1 : (A1)-(A2) IMPLY (B1)-(B2)

Assume that (A1) and (A2) hold true. Then, let $f_G : \mathcal{M}^+(\Omega) \to \mathcal{M}^+(\mathbb{R}^{d'})$ be the map, $f_G(\mu) = G(\mu)_{\#}\mu$, with $G : \mathcal{M}^+(\Omega) \times \Omega \to \mathbb{R}^{d'}$. It is straightforward to prove (B1) and, hence, we will focus on proving that (B2) holds. We assume that $\psi_k, \psi \in C_0^1(\mathbb{R}^{d'})$ and $\mu_k, \mu \in \mathcal{M}_+(\Omega)$, $k = 1, 2, \ldots$ are sequences with

$$\psi_k \text{ is constant in an open neighborhood of supp}(f(\mu_k))$$

and
$$\lim_{k\to\infty} W_1(\mu_k, \mu) = 0, \quad \lim_{k\to\infty} \|\psi_k - \psi\|_{L^\infty(\mathbb{R}^{d'})} = 0.$$

Let
$$\mu_{k,x}^\epsilon := \mu_k + \epsilon \delta_x.$$

Then
$$
\begin{aligned}
\langle f_G(\mu_{k,x}^\epsilon), \psi \rangle &= \langle (G(\mu_{k,x}^\epsilon))_\# \mu_{k,x}^\epsilon, \psi_k \rangle \\
&= \langle \mu_{k,x}^\epsilon, \psi_k \circ G(\mu_{k,x}^\epsilon, \cdot) \rangle \\
&= \int_{\mathbb{R}^d} \psi_k(G(\mu_{k,x}^\epsilon, y)) d\mu_{k,x}^\epsilon(y) \\
&= \int_{\mathbb{R}^d} \psi_k(G(\mu_{k,x}^\epsilon, y)) d\mu_k(y) + \epsilon \psi_k(G(\mu_{k,x}^\epsilon, x)).
\end{aligned}
$$

By (A2), $(\mu, x) \to G(\mu, x)$ is a continuous function. As the set $\Omega \subset \mathbb{R}^d$ is compact, for any $\rho > 0$ the set
$$\mathcal{M}_\rho^+(\Omega) := \{\mu \in \mathcal{M}^+(\Omega) : \mu(\Omega) \leq \rho\},$$
consists of measures that are uniformly bounded and supported in the same compact set $\Omega$. Therefore, the set $\mathcal{M}_\rho^+(\Omega)$ is tight (see (Billingsley, 1999, Chapter 1, Section 1)). The set $\mathcal{M}_\rho^+(\Omega)$ is also closed in the weak$^*$ topology of measures as it is closed in the 1-Wasserstein topology. By Prokhorov's theorem (see (Billingsley, 1999, Chapter 5, Theorem 5.1)), the set $\mathcal{M}_\rho^+(\Omega)$ is compact in the 1-Wasserstein topology. As a continuous map defined in a compact metric space is uniformly continuous, the map $G : \mathcal{M}_\rho^+(\Omega) \times \Omega \to \mathbb{R}^{d'}$ is uniformly continuous. Moreover, by our assumptions, the derivative of $\psi_k$ is zero in some neighborhood, $V \subset \mathbb{R}^{d'}$, of the finite set $\{G(\mu_k, x) : x \in \mathrm{supp}\,(\mu_k)\}$. As $G(\mu_{k,x}^\epsilon, \cdot) \to G(\mu_k, \cdot)$ uniformly in $\Omega \subset \mathbb{R}^d$ as $\epsilon \to 0$, we see that
$$\sup_{y \in \mathrm{supp}(\mu_k)} |G(\mu_{k,x}^\epsilon, y) - G(\mu_k, y)| \to 0 \quad \text{as } \epsilon \to 0. \tag{40}$$

Thus, we find that for sufficiently small $\epsilon \in (0, 1)$, for all $y \in \mathrm{supp}\,(\mu_k)$ the point $G(\mu_{k,x}^\epsilon, y)$ belongs to the set $V$, and, hence,
$$\psi_k(G(\mu_{k,x}^\epsilon, y)) = \psi_k(G(\mu_k, y))$$
for all $y \in \mathrm{supp}\,(\mu_k)$, and
$$\langle f_G(\mu_{k,x}^\epsilon), \psi_k \rangle = \int_{\mathbb{R}^d} \psi_k(G(\mu_k, y)) d\mu_k(y) + \epsilon \psi_k(G(\mu_k, x)).$$

This implies that
$$\left.\frac{d}{d\epsilon}\right|_{\epsilon=0} \langle f_G(\mu_{k,x}^\epsilon), \psi_k \rangle = 0 + \psi_k(G(\mu_k, y)) = \psi_k(G(\mu_k, y)). \tag{41}$$

Thus,
$$\overline{\mathcal{D}}_{f_G}(\mu_k, x, \psi_k) = \psi_k(G(\mu_k, x)).$$

We see that
$$\lim_{k\to\infty} |\psi_k(G(\mu_k, x)) - \psi_k(G(\mu_k, x))| \leq \lim_{n\to\infty} \|\psi_k - \psi\|_{L^\infty(\mathbb{R}^{d'})} = 0,$$
and as $\mu \to G(\mu, x)$ is continuous, we have
$$\lim_{k\to\infty} \psi(G(\mu_k, x)) = \psi(G(\mu, x)).$$

But then
$$\lim_{k\to\infty} \psi_k(G(\mu_k, x)) = \psi(G(\mu, x)).$$

As the above holds for all $\mu_k$ and $\psi_k$ that converge to $\mu$ and $\psi$ in the way stated in Definition 1, we conclude that the regular part of the derivative $\overline{\mathcal{D}}_{f_G}(\mu, y, \psi)$ exists and is equal to
$$\overline{\mathcal{D}}_{f_G}(\mu, x, \psi) = \psi(G(\mu, y)). \tag{42}$$

This proves the existence of the regular part of the derivative. Moreover, these arguments prove Lemma 3.

Next, we prove that the map $(\mu, y, \psi) \to \overline{\mathcal{D}}_{f_G}(\mu, y, \psi)$ is uniformly continuous when (A1) and (A2) are valid. To this end, let $\epsilon_1 > 0$. By (A2), there is a $\delta_1 = \delta_1(\epsilon_1) \in (0, \epsilon_1)$ such that if $W_1(\mu_1, \mu_2) < \delta_1(\epsilon_1)$ and $|y_1 - y_2| < \delta_1(\epsilon_1)$ then $|G(\mu_1, y_1) - G(\mu_2, y_2)| < \epsilon_1/2$. Let $(\mu_1, y_1, \psi_1), (\mu_2, y_2, \psi_2) \in \mathcal{X}$ so that $\mathrm{Lip}(\psi_j) \leq \eta$ for $j = 1, 2$. Also, assume that $\|\psi_1 - \psi_2\|_{L^\infty} < \delta_1(\epsilon_1)$. Equation (42) implies that

$$
\begin{aligned}
|\overline{\mathcal{D}}_{f_G}(\mu_1, y_1, \psi_1) - \overline{\mathcal{D}}_{f_G}(\mu_2, y_2, \psi_2)| &= |\psi_1(G(\mu_1, y_1)) - \psi_2(G(\mu_2, y_2))| \\
&\leq |\psi_1(G(\mu_1, y_1)) - \psi_1(G(\mu_2, y_2))| \\
&\quad + |\psi_1(G(\mu_2, y_2)) - \psi_2(G(\mu_2, y_2))| \\
&\leq \mathrm{Lip}(\psi_1)|G(\mu_1, y_1) - G(\mu_2, y_2)| + \|\psi_1 - \psi_2\|_{L^\infty} \\
&\leq \mathrm{Lip}(\psi_1)\epsilon_1 + \delta_1(\epsilon_1) \\
&\leq (\eta + 1)\epsilon_1.
\end{aligned}
$$

We observe that if $D_{\mathcal{X}}((\mu_1, y_1, \psi_1), (\mu_2, y_2, \psi_2)) < \delta_1(\epsilon_1)$ then $W_1(\mu_1, \mu_2) < \delta_1(\epsilon_1)$ and $|y_1 - y_2| < \delta_1(\epsilon_1)$, and moreover that $\|\psi_1 - \psi_2\|_{L^\infty} < \delta_1(\epsilon_1)$. We conclude that $(\mu, y, \psi) \to \overline{\mathcal{D}}_{f_G}(\mu, y, \psi)$ is uniformly continuous. This proves (B2).

We continue with proving one direction of the final statement of the theorem. Let $G(\mu, y)$ be a Lipschitz map. Equation (42) implies that for all $(\mu_1, y_1, \psi_1), (\mu_2, y_2, \psi_2) \in \mathcal{X}$,

$$
\begin{aligned}
|\overline{\mathcal{D}}_{f_G}(\mu_1, y_1, \psi_1) - \overline{\mathcal{D}}_{f_G}(\mu_2, y_2, \psi_2)| &= |\psi_1(G(\mu_1, y_1)) - \psi_2(G(\mu_2, y_2))| \\
&\leq |\psi_1(G(\mu_1, y_1)) - \psi_1(G(\mu_2, y_2))| \\
&\quad + |\psi_1(G(\mu_2, y_2)) - \psi_2(G(\mu_2, y_2))| \\
&\leq \mathrm{Lip}(\psi_1)|G(\mu_1, y_1) - G(\mu_2, y_2)| + \|\psi_1 - \psi_2\|_{L^\infty} \\
&\leq (\mathrm{Lip}(\psi_1)\mathrm{Lip}(G) + 1)D_{\mathcal{X}}((\mu_1, y_1, \psi_1), (\mu_2, y_2, \psi_2)) \\
&\leq (\eta\mathrm{Lip}(G) + 1)D_{\mathcal{X}}((\mu_1, y_1, \psi_1), (\mu_2, y_2, \psi_2)).
\end{aligned}
$$

Hence, $(\mu, y, \psi) \to \overline{\mathcal{D}}_{f_G}(\mu, y, \psi)$ is a Lipschitz map.

### C.2 PART 2 : (B1)-(B2) IMPLY (A1)-(A2)

Assume that (B1) and (B2) hold true. Since $f$ is a support-preserving map, $f : \mathcal{M}^+(\Omega) \to \mathcal{M}^+(\mathbb{R}^{d'})$, there are (possibly non-continuous) functions,

$$
y_i : \Omega^n \times (0, \infty)^n \to \mathbb{R}^{d'}, \ (\boldsymbol{x}, \boldsymbol{a}) \to y_i(\boldsymbol{x}; \boldsymbol{a}), \quad i = 1, 2, \ldots, n,
$$

where

$$
\boldsymbol{x} = (x_1, \ldots, x_n) \quad \text{and} \quad \boldsymbol{a} = (a_1, \ldots, a_n),
$$

such that the following holds: Let

$$
\mu = \sum_{i=1}^n a_i \delta_{x_i} \in \mathcal{M}_{fin}(\Omega), \quad a_i > 0;
$$

then the functions $y_i(\boldsymbol{x}; \boldsymbol{a})$ satisfy

$$
f(\mu) = \sum_{i=1}^n a_i \delta_{y_i(\boldsymbol{x};\boldsymbol{a})}.
$$

When $\mu \in \mathcal{M}^+_{fin,dif,(n)}(\Omega)$ (which is a refinement of the property that if $j \neq i$ then $a_j \neq a_i$), the functions $(\boldsymbol{x}; \boldsymbol{a}) \to y_i(\boldsymbol{x}; \boldsymbol{a})$ must have the following property,

$$
\text{if } x_j = x_i \text{ then } y_j(\boldsymbol{x}; \boldsymbol{a}) = y_i(\boldsymbol{x}; \boldsymbol{a}). \tag{43}
$$

Let $\mu \in \mathcal{M}^+(\Omega)$ and $x \in \Omega$, and $\alpha \in C_0^\infty(\mathbb{R}^d)$ be a cutoff function such that $\alpha(x) = 1$ for all $x \in \Omega$ and $\mathrm{Lip}(\alpha(x) \cdot x) \leq \eta$. We define

$$
G(\mu, x) := \begin{pmatrix} \overline{\mathcal{D}}_f(\mu, x, \alpha\pi_1) \\ \vdots \\ \overline{\mathcal{D}}_f(\mu, x, \alpha\pi_{d'}) \end{pmatrix}, \tag{44}
$$

where $\pi_\ell : \mathbb{R}^d \to \mathbb{R}$ is the projection $\pi_\ell(x) = x_\ell$ onto the $\ell$-th component. By (B2), the map $(\mu, x) \mapsto G(\mu, x)$ is continuous, which proves (A2). In what follows, we will prove (A1).

**The case when $\mu \in \mathcal{M}^+_{fin,dif,(n)}(\Omega)$.** We let

$$\mu = \sum_{i=1}^n a_i \delta_{x_i} \in \mathcal{M}^+_{fin,dif,(n)}(\Omega)$$

and

$$f(\mu) = \sum_{i=1}^n a_i \delta_{y_i(\boldsymbol{x};\boldsymbol{a})}.$$

We define the measures,

$$\mu_x^\epsilon = \epsilon \delta_x + \sum_{i=1}^n a_i \delta_{x_i}.$$

We observe that when $\epsilon > 0$ is small enough, it holds that $\mu_x^\epsilon \in \mathcal{M}^+_{fin,dif,(n)}(\Omega)$ if $x \in \{x_1, \ldots, x_n\}$, or $\mu_x^\epsilon \in \mathcal{M}^+_{fin,dif,(n+1)}(\Omega)$ if $x \notin \{x_1, \ldots, x_n\}$.

With the notation, $(\boldsymbol{x}, x) = (x_1, \ldots, x_n, x)$, $(\boldsymbol{a}, \epsilon) = (a_1, \ldots, a_n, \epsilon)$ and sometimes indicating the number, $n$ say, of variables in the function $y_i$ as $y_i^{(n)}$, we find that

$$f(\mu_x^\epsilon) = \epsilon \delta_{y_{n+1}^{(n+1)}(\boldsymbol{x},x;\boldsymbol{a},\epsilon)} + \sum_{i=1}^n a_i \delta_{y_i^{(n+1)}(\boldsymbol{x},x;\boldsymbol{a},\epsilon)}. \tag{45}$$

We consider the case when $x = x_j$. Then,

$$\left( \sum_{i=1}^n a_i \delta_{x_i} + \epsilon \delta_x \right) \Big|_{x=x_j} = \sum_{i \in \{1,\ldots,n\}\setminus\{j\}} a_i \delta_{x_i} + (a_j + \epsilon) \delta_{x_j} \in \mathcal{M}^+_{fin,(n)}(\mathbb{R}). \tag{46}$$

Thus, when we write $x = x_{n+1} = x_j$, it holds that

$$y_{n+1}^{(n+1)}(\boldsymbol{x}, x; \boldsymbol{a}, \epsilon) \Big|_{x=x_j} = y_{n+1}^{(n+1)}(\boldsymbol{x}, x_{n+1}; \boldsymbol{a}, \epsilon) \Big|_{x_{n+1}=x_j} = y_j^{(n)}(\boldsymbol{x}; \boldsymbol{a} + \epsilon e_j), \tag{47}$$

where $e_j = (0, 0, \ldots, 0, 1, 0, \ldots, 0) = (\delta_{ij})_{i=1}^n$, whence

$$\boldsymbol{a} + \epsilon e_j = (a_1, \ldots, a_{j-1}, a_j + \epsilon, a_{j+1}, \ldots, a_n).$$

By Lemma 4, and using equations (46)-(47), we arrive at

$$\lim_{\epsilon \to 0+} y_{n+1}^{(n+1)}(\boldsymbol{x}, x; \boldsymbol{a}, \epsilon) \Big|_{x=x_j} = y_j^{(n)}(\boldsymbol{x}; \boldsymbol{a}). \tag{48}$$

Let $\ell \in [d']$. We choose $\psi_k^{(\ell)} \in C_0^1(\mathbb{R}^{d'})$ such that

$$\psi_k^{(\ell)} \quad \text{is constant in an open neighborhood of supp}\,(f(\mu))$$

and

$$\mathrm{Lip}(\psi_k^{(\ell)}) \le \eta \text{ together with } \lim_{n \to \infty} \|\psi_k^{(\ell)} - \alpha \pi_\ell\|_{L^\infty(\mathbb{R}^{d'})} = 0.$$

Thus, by using equation (48), we obtain

$$\begin{aligned}
\overline{\mathcal{D}}_f(\mu, x, \psi_k^{(\ell)}) \Big|_{x=x_j} &= \overline{\mathcal{D}}_f(\mu, x_{n+1}, \psi_k^{(\ell)}) \Big|_{x_{n+1}=x_j} \\
&= \lim_{\epsilon \to 0+} \langle \psi_k^{(\ell)}, \delta_{y_{n+1}^{(n+1)}(\boldsymbol{x},x_{n+1};\boldsymbol{a},\epsilon)} \rangle \Big|_{x_{n+1}=x_j} \\
&= \lim_{\epsilon \to 0+} \psi_k^{(\ell)}(y_{n+1}^{(n+1)}(\boldsymbol{x}, x_{n+1}; \boldsymbol{a}, \epsilon)) \Big|_{x_{n+1}=x_j} \\
&= \lim_{\epsilon \to 0+} \psi_k^{(\ell)}(y_j^{(n)}(\boldsymbol{x}; \boldsymbol{a} + \epsilon e_j)) \\
&= \psi_k^{(\ell)}(y_j^{(n)}(\boldsymbol{x}; \boldsymbol{a})) = \psi_k^{(\ell)}(y_j(\boldsymbol{x}; \boldsymbol{a})).
\end{aligned}$$

By the definition of $\overline{\mathcal{D}}_f(\mu, x, \alpha\pi_\ell)$ and that $\psi_k^{(\ell)} \to \alpha\pi_\ell$ in $L^\infty(\mathbb{R}^{d'})$ as $n \to \infty$, we observe that for $x = x_j$,

$$
\begin{aligned}
\pi_\ell(G(\mu, x_j)) &= \overline{\mathcal{D}}_f(\mu, x_j, \alpha\pi_\ell) = \lim_{n\to\infty} \overline{\mathcal{D}}_f(\mu, x_j, \psi_k^{(\ell)}) \\
&= \lim_{n\to\infty} \psi_k^{(\ell)}(y_j(\boldsymbol{x}; \boldsymbol{a})) = (\alpha\pi_\ell)(y_j(\boldsymbol{x}; \boldsymbol{a})) = \pi_\ell(y_j).
\end{aligned}
$$

This proves that, for each $j \in [n]$,

$$
G(\mu, x_j) = y_j(\boldsymbol{x}; \boldsymbol{a}), \tag{49}
$$

which is equivalent to

$$
f(\mu) = (G_\mu)_\# \mu \quad \text{for } \mu \in \mathcal{M}^+_{fin,dif,(n)}(\Omega).
$$

**The case when** $\mu \in \mathcal{M}^+(\Omega)$**.** Let $\mu$ be a (possibly not-discretely supported) measure $\mu \in \mathcal{M}^+(\Omega)$. By Lemma 1, we choose the sequence $(\tilde{\mu}_k)_{k\in\mathbb{N}} \subset \mathcal{M}^+_{fin,dif}(\Omega)$ such that $\tilde{\mu}_k \to \mu$ as $k \to \infty$, where the limit is considered in the 1-Wasserstein topology. We have already shown that for $\tilde{\mu}_k \in \mathcal{M}^+_{fin,dif}(\Omega)$,

$$
f(\tilde{\mu}_k) = (G_{\tilde{\mu}_k})_\#(\tilde{\mu}_k).
$$

Hence, taking the limit,

$$
f(\mu) = \lim_{m\to\infty} (G_{\tilde{\mu}_k})_\#(\tilde{\mu}_k). \tag{50}
$$

That is, for all $\psi \in C_0^1(\mathbb{R}^{d'})$,

$$
\langle \psi, f(\mu) \rangle = \lim_{m\to\infty} \langle \psi, (G_{\tilde{\mu}_k})_\#(\tilde{\mu}_k) \rangle, \tag{51}
$$

where

$$
\langle \psi, (G_{\tilde{\mu}_k})_\#(\tilde{\mu}_k) \rangle = \langle \psi \circ G_{\tilde{\mu}_k}, \tilde{\mu}_k \rangle = \int_{\mathbb{R}^{d'}} \psi(G_{\tilde{\mu}_k}(x)) \, d\tilde{\mu}_k(x).
$$

But then

$$
\begin{aligned}
\lim_{k\to\infty} \langle \psi, (G_{\tilde{\mu}_k})_\#(\tilde{\mu}_k) \rangle &= \lim_{k\to\infty} \int_{\mathbb{R}} \psi(G_{\tilde{\mu}_k}(x)) \, d\tilde{\mu}_k(x) \\
&= \lim_{k\to\infty} \int_{\mathbb{R}} (\psi(G(\tilde{\mu}_k, x)) - \psi(G(\mu, x))) \, d\tilde{\mu}_k(x) + \int_{\mathbb{R}} \psi(G(\mu, x)) \, d\tilde{\mu}_k(x). \quad (52)
\end{aligned}
$$

By condition (B2), $(\mu, x) \to G(\mu, x)$ is uniformly continuous so that, using the compactness of $\Omega$,

$$
\|\psi(G(\tilde{\mu}_k, \cdot)) - \psi(G(\mu, \cdot))\|_{L^\infty(\Omega)} \le \|\psi\|_{C^1} \|G(\tilde{\mu}_k, \cdot) - G(\mu, \cdot)\|_{L^\infty(\Omega)} \to 0 \text{ as } k \to \infty.
$$

Hence, equations (51) and (52) imply that

$$
\begin{aligned}
\langle \psi, f(\mu) \rangle &= 0 + \lim_{k\to\infty} \int_{\mathbb{R}} \psi(G(\mu, x)) \, d\tilde{\mu}_k(x) \\
&= \lim_{k\to\infty} \langle \psi(G(\mu, \cdot)), \tilde{\mu}_k \rangle = \langle \psi(G(\mu, \cdot)), \mu \rangle = \langle \psi, (G_\mu)_\# \mu \rangle \quad (53)
\end{aligned}
$$

for all $\psi \in C_0^1(\mathbb{R}^{d'})$, $\mathrm{Lip}(\psi) \le \eta$. As both sides of (53) are linear in $\psi$, we see that (53) holds for all $\psi \in C_0^1(\mathbb{R}^{d'})$ and, therefore, for all $\psi \in C_0(\mathbb{R}^{d'})$. Thus,

$$
f(\mu) = (G_\mu)_\# \mu \quad \text{for } \mu \in \mathcal{M}^+(\Omega).
$$

This implies (A1).

Finally, we observe that if $(\mu, y, \psi) \to \overline{\mathcal{D}}_{f_G}(\mu, y, \psi)$ is a Lipschitz map then $(\mu, y) \to G(\mu, y) = (\overline{\mathcal{D}}_f(\mu, y, \alpha\pi_j))_{j=1}^{d'}$ is also Lipschitz.

# D  THE REGULAR PART OF THE DERIVATIVE

We provide some perspectives on the regular part of the derivative introduced in the main text, in the following remarks.

**Remark 1.** *A similar situation occurs when one defines the generalization of a derivative for a Lipschitz function $h : \mathbb{R}^d \to \mathbb{R}$. By the Rademacher theorem, the classical derivative of $h$ exists outside a zero-measurable set; to overcome this, one defines a weak derivative that is a function in $L^1_{loc}(\mathbb{R}^d)$ and is defined almost everywhere. We recall that the weak derivative is defined, in the sense of distributions, by the formula*

$$\langle \partial_{x_i} h, \psi \rangle = - \int_{\mathbb{R}^d} h(x) \partial_{x_i} \psi(x) dd, \quad for \; \psi \in C_0^\infty(\mathbb{R}^d).$$

*In the case when $h$ is a $C^1$-function, the classical derivative coincides with the weak derivative and the distributional duality coincides with the $L^2$-inner product*

$$\langle \partial_{x_i} h, \psi \rangle = \int_{\mathbb{R}^d} \partial_{x_i} h(x) \psi(x) dx.$$

*In this setting, the weak derivative is defined for a larger class of functions as a "new" generalized function.*

*Our definition of the regular part of the derivative is defined as a new generalized function using duality (or, in the weak sense). This definition is formally quite different from the classical one of Fréchet derivative. However, as we see in Lemma 3, for map $f_G$ defined with a smooth in-context function $G$, the regular part of derivative $\overline{\mathcal{D}}_{f_G}(\mu, x, \psi)$ coincides with the above defined object, $D^{reg}_{f_G}(\mu, x, \psi)$. So, we consider $D^{reg}_{f_G}(\mu, x, \psi)$ as a new object that is different from the classical Fréchet derivative, and show that the definition of $D^{reg}_{f_G}(\mu, x, \psi)$ can be extended as a generalized regular part of the derivative, $\overline{\mathcal{D}}_f(\mu, x, \psi)$, for a class of functions $f$ for which we do not assume that the classical Fréchet derivative is well-defined.*

**Remark 2.** *We point out that for any support preserving map $f : \mathcal{M}^+(\Omega) \to \mathcal{M}^+(\mathbb{R}^{d'})$, $\mu \in \mathcal{M}^+(\Omega)$, and $\psi \in C_0^1(\mathbb{R}^{d'})$, we can find a sequence of finitely supported measures, $\mu_k \in \mathcal{M}^+_{fin}(\Omega)$, that converges in the 1-Wasserstein topology to $\mu$ as $n \to \infty$. Then also $supp\,(f(\mu_k))$ is finitely supported, and we can denote $supp\,(f(\mu_k)) = \{y_{1,n}, y_{2,n}, \ldots, y_{m_n,n}\}$. We can modify the function $\psi$ in a small neighborhood of each point, $y_{j,n}$, so that we obtain a function $\psi_k \in C_0^1(\mathbb{R}^{d'})$ that satisfies (7) and $\psi_k$ converges in the $L^\infty$ topology to $\psi$ as $n \to \infty$. Thus, we see that for all measures $\mu \in \mathcal{M}^+(\Omega)$ and $\psi \in C_0^1(\mathbb{R}^{d'})$, we can find sequences $\mu_k$ and $\psi_k$ that satisfy the conditions in Definition 4. The existence of $\overline{\mathcal{D}}_f(\mu, x, \psi)$ thus means that for all $\mu$, $x$, and $\psi$, the limits (6) exist and are independent of the chosen sequences $\mu_k$ and $\psi_k$.*

**Remark 3.** *We can come up with an alternative version of the definition for the regular part of the Fréchet derivative of $f$ and of Definition 4. Let us consider the triplets of measures $\mu$, points $x$ and test functions $\psi$ having the property that the test functions are locally constant in the support of $f(\mu)$. We denote this set by*

$$\mathcal{P}_{lc} = \mathcal{P}_{lc}(f, \Omega, \rho, \eta) = \{(\mu, x, \psi) \in \mathcal{M}^+(\Omega) \times \Omega \times C_0^1(\mathbb{R}^{d'}) :$$
$$\mu(\Omega) \leq \rho, \;\; \psi \; is \; constant \; in \; an \; open \; neighborhood \; of \; f(\mu), \;\; \|\psi\|_{C^1} \leq \eta\}.$$

*Let*
$$\mathcal{L}_f(\mu, x, \psi) = \langle \psi, D_\mu f(\mu)[\delta_x] \rangle,$$
*be the duality of the Fréchet derivative $D_\mu f(\mu)[\delta_x]$ and the test function $\psi$. Then the restriction of $\mathcal{L}_f$ to the set $\mathcal{P}_{lc}$, that is,*
$$\mathcal{L}_f|_{\mathcal{P}_{lc}} : \mathcal{P}_{lc} \to \mathbb{R},$$
*coincides with the regular part of the derivative $\overline{\mathcal{D}}_f(\mu, x, \psi)$ of $f$. When the regular part of the derivative of $f$ exists, this map has a continuous extension to the set $\mathcal{X}$,*
$$\mathcal{L}_f^{ext} : \mathcal{X} \to \mathbb{R},$$
*in the topology determined by the metric, $D_\mathcal{X}$. This extension is the map $(\mu, x, \psi) \to \overline{\mathcal{D}}_f(\mu, x, \psi)$. Hence, $\overline{\mathcal{D}}_f(\mu, x, \psi)$ given in Definition 4 can also be defined as the extension of the usual Fréchet derivative from the set $\mathcal{P}_{lc}$ to the completion of this set in the appropriate topology.*

# E  MLPs WITH SKIP CONNECTIONS AND COMPOSITION FORMING AN IN-CONTEXT MAP

We consider MLPs with possible skip connections, denoted by $F_\eta$, that are given by the function

$$F_\eta : \mathbb{R}^d \to \mathbb{R}^d, \quad F_\eta = c_\eta \cdot Id_x + \sigma \circ (A_\eta^L + b_\eta^L) \circ \cdots \circ \sigma \circ (A_\eta^1 + b_\eta^1), \tag{54}$$

where $c_\eta \in \mathbb{R}$, $A_\eta^j \in \mathbb{R}^{d_j \times d_{j-1}}$ are the weight matrices, $b_\eta^j \in \mathbb{R}^{d_j}$ are bias vectors, $\sigma$ is an activation function, for example the sigmoid function, and $d_0 = d_L = d$. This defines a map for measures, $f_{F_\eta} = (F_\eta)_\# : \mathcal{M}^+(\mathbb{R}^d) \to \mathcal{M}^+(\mathbb{R}^d)$ that for discrete measure $\nu = \sum_{i=1}^n \frac{1}{n} \delta_{y_i}$ is given by

$$f_{F_\eta}(\nu) = (F_\eta)_\# \left( \sum_{i=1}^n \frac{1}{n} \delta_{y_i} \right) = \sum_{i=1}^n \frac{1}{n} \delta_{z_i} \tag{55}$$

where

$$z_i = F_\eta(y_i). \tag{56}$$

The composition $f_{F_\eta} \diamond f_{\Gamma_\xi} : \mathcal{M}^+(\Omega) \to \mathcal{M}^+(\mathbb{R}^d)$ of the maps $f_{F_\eta}$ and $f_{\Gamma_\xi}$, see (8), maps the discrete measure $\mu$, given in (21), to

$$(f_{F_\eta} \diamond f_{\Gamma_\xi}) \left( \sum_{i=1}^n \frac{1}{n} \delta_{x_i} \right) = \sum_{i=1}^n \frac{1}{n} \delta_{z_i}, \quad z_i = F_\eta(\Gamma_\xi(\mu, x_i)). \tag{57}$$

We write

$$H_\eta(x) := F_\eta(x) - x.$$

We note that as $\Gamma_\xi(\mu, x) = x + \text{Att}_\xi(\mu, x)$ and $F_\eta(x) = x + H_\eta(x)$, we can write

$$F_\eta(\Gamma_\xi(\mu, x)) = x + \mathcal{V}(\mu, x) \tag{58}$$

and

$$f_{F_\eta} \diamond f_{\Gamma_\xi} = Id_x + f_{\mathcal{V}}, \tag{59}$$

where $\mathcal{V} : \mathcal{M}^+(\Omega) \times \mathbb{R}^d \to \mathbb{R}^d$ is the map $\mathcal{V} = \text{Att}_\xi + H_\eta \circ \Gamma_\xi$, that is,

$$\mathcal{V}(\mu, x) = \text{Att}_\xi(\mu, x) + H_\eta(\Gamma_\xi(\mu, x)) = \text{Att}_\xi(\mu, x) + H_\eta \circ (Id_x + \text{Att}_\xi(\mu, \cdot))(x). \tag{60}$$

# F  A COUNTEREXAMPLE FOR THE CHARACTERIZATION OF SUPPORT-PRESERVING MAPS USING ONLY CONTINUITY

In this section, we construct a map $f : \mathcal{P}(\Omega) \to \mathcal{P}(\Omega)$ that is support preserving and continuous in the 1-Wasserstein topology, but which cannot be represented as $f_G$ using a continuous in-context map, $G$. Such a map, $f$, is given in formulas (63)-(65) below. This shows the importance of the assumptions on the derivative of the map $f$ in the main theorem.

Let us next prove Proposition 1. We recall the statement:

**Proposition 3.** *Let $d = 1$ and $\Omega = [-3, 3] \subset \mathbb{R}$ and consider the set $\mathcal{P}(\Omega)$ endowed with the 1-Wasserstein topology. There exists a continuous, support preserving map $f : \mathcal{P}(\Omega) \to \mathcal{P}(\Omega)$ such that there does not exist a continuous map $G : \mathcal{P}(\Omega) \times \Omega \to \Omega$ for which $f = f_G$.*

*Proof.* For $0 \le a \le 1$, we define

$$R_a : [-3, 3] \to [-3, 3], \tag{61}$$

$$R_a(x) = \begin{cases} x, & \text{for } x \le -1 \text{ or } x \ge 1, \\ x + \frac{1}{10} \cos^2(\frac{1}{2}\pi x) \cos(ax), & \text{for } -1 < x < 1. \end{cases}$$

We note that the derivative of $R_a$ is given by

$$R_a' : [-3, 3] \to \mathbb{R}, \tag{62}$$

$$R_a'(x) = \begin{cases} 0, & \text{for } x \le -1 \text{ or } x \ge 1, \\ 1 - \frac{\pi}{10} \cos(\frac{1}{2}\pi x) \sin(\frac{1}{2}\pi x) \cos(ax) - \frac{a}{10} \cos^2(\frac{1}{2}\pi x) \sin(ax), & \text{for } -1 < x < 1 \end{cases}$$

and that $R_a : [-3, 3] \to [-3, 3]$ is a $C^1$ function that maps $R_a : [-3, 3] \to [-3, 3]$. Moreover, we point out that when $a = 0$, $R_0(x) = x$.

Next, we consider the map

$$f(\mu) = (R_{a(\mu)})_\# \mu, \tag{63}$$

where

$$a(\mu) = \begin{cases} \frac{1}{\kappa(\mu)} & \text{if } \kappa(\mu) > 0, \\ 0 & \text{if } \kappa(\mu) = 0 \end{cases} \tag{64}$$

and

$$\kappa(\mu) = \int_{-2}^{-1} (2 - |x|) d\mu(x) + \int_{-1}^{1} d\mu(x) + \int_{1}^{2} (2 - x) d\mu(x). \tag{65}$$

The function $\mu \to \kappa(\mu)$ is a continuous map $\mathcal{P}([-3, 3]) \to \mathbb{R}$ but $\mu \to a(\mu)$ is not continuous.

By Vallender (1974), the 1-Wasserstein distance satisfies,

$$W_1(\mu_1, \mu_2) = \int_{[-3,3]} |F_1(x) - F_2(x)| \, \mathrm{d}x, \tag{66}$$

where $F_1(x) = \mu_1([-3, x])$ and $F_2(x) = \mu_2([-3, x])$ are the cumulative distribution functions of $\mu_1$ and $\mu_2$, respectively. Thus, as $R_{a(\mu)}(x)$ is the identity map for $x \in [-3, 3] \setminus [-\frac{3}{2}, \frac{3}{2}]$ and $R_{a(\kappa)}$ maps the interval $[-\frac{3}{2}, \frac{3}{2}]$ to itself, we find that

$$W_1(f(\mu_1), f(\mu_2)) \leq \text{diam}\left(\left[-\frac{3}{2}, \frac{3}{2}\right]\right)\left|\mu_1\left(\left[-\frac{3}{2}, \frac{3}{2}\right]\right) - \mu_2\left(\left[-\frac{3}{2}, \frac{3}{2}\right]\right)\right| + W_1(\mu_1, \mu_2)$$

$$\leq 3\left(\mu_1\left(\left[-\frac{3}{2}, \frac{3}{2}\right]\right) + \mu_2\left(\left[-\frac{3}{2}, \frac{3}{2}\right]\right)\right) + W_1(\mu_1, \mu_2). \tag{67}$$

**Lemma 5.** *The map, $f : \mathcal{P}([-3, 3]) \to \mathcal{P}([-3, 3])$, is continuous in the 1-Wasserstein topology and is a support-preserving map.*

*Proof.* When $\nu = \sum_{i=1}^{n} \frac{1}{n} \delta_{x_i}$, we have by the definition of $f$ (cf. (63)) that

$$f(\nu) = (R_{a_0})_\# \nu,$$

where $a_0 = a(\nu)$. As $R_{a_0}$ is a $C^1$-map, we see that

$$f(\nu) = \sum_{i=1}^{n} \frac{1}{n} \delta_{y_i}, \quad y_i = R_{a_0}(x_i). \tag{68}$$

This shows that $f$ is a support-preserving map.

Let $\mu_k, \mu \in \mathcal{P}([-3, 3])$ satisfy

$$\lim_{k \to \infty} W_1(\mu_k, \mu) = 0. \tag{69}$$

We will next show that

$$\lim_{k \to \infty} W_1(f(\mu_k), f(\mu)) = 0. \tag{70}$$

First, we consider the case when $\kappa(\mu) > 0$. In this case, also $\kappa(\mu_k) > 0$ when $n$ is large enough. Then, we can use the fact that $(x, a) \to R_a(x)$ is $C^1$-smooth in the domain $(x, a) \in [-3, 3] \times (0, 1]$, i.e., when $a$ is strictly positive. This implies that the limit (70) is valid when $\kappa(\mu) > 0$.

Second, we consider the case when $\kappa(\mu) = 0$. Then, $\mu([-\frac{3}{2} - \frac{1}{10}, \frac{3}{2} + \frac{1}{10}]) = 0$ and $f(\mu) = \mu$. For all $\epsilon > 0$ there is $n_\epsilon > 0$ such that for $n \geq n_\epsilon$ it holds that $W_1(\mu_n, \mu) < \epsilon$ and $\mu_k([-\frac{3}{2}, \frac{3}{2}]) < \epsilon$.

We see that $R_{a(\mu)}(x)$ is the identity map for $x \in [-3,3] \setminus [-\frac{3}{2}, \frac{3}{2}]$ and $R_{a(\kappa)}$ maps the interval $[-\frac{3}{2}, \frac{3}{2}]$ to itself. Thus, $n \geq n_\epsilon$, we have by (67),

$$
\begin{aligned}
W_1(f(\mu_k), f(\mu)) &\leq 3\left(\mu_k\left(\left[-\frac{3}{2}, \frac{3}{2}\right]\right) + \mu\left(\left[-\frac{3}{2}, \frac{3}{2}\right]\right)\right) + W_1(\mu_k, \mu) \\
&\leq 3\epsilon + W_1(\mu_k, \mu) \\
&\leq 4\epsilon.
\end{aligned}
\tag{71}
$$

These show that the limit (70) is valid also when $\kappa(\mu) > 0$. This proves that the limit (70) is valid. Hence, $f$ is continuous in 1-Wassestein metric. This proves the claim. $\square$

In the following, we use the 1-Wasserstein topology in the set $\mathcal{P}([-3,3])$.

**Lemma 6.** *There are no continuous maps $G : \mathcal{P}([-3,3]) \times [-3,3] \to [-3,3]$, such that*

$$
f(\mu) = f_G(\mu).
\tag{72}
$$

*Proof.* For $\epsilon > 0$, let

$$
\begin{aligned}
\mu_\epsilon &= (1-\epsilon)\delta_{x_0} + \epsilon\delta_{\sqrt{\epsilon}}, \\
\nu_\epsilon &= (1-\epsilon)\delta_{x_0} + \epsilon\delta_{R_{1/\epsilon}(\sqrt{\epsilon})},
\end{aligned}
$$

where $x_0 = 2$. We see that as $\epsilon \to 0$, we have

$$
\lim_{\epsilon \to 0} W_1(\mu_\epsilon, \delta_{x_0}) = 0,
\tag{73}
$$

$$
\lim_{\epsilon \to 0} W_1(\nu_\epsilon, \delta_{x_0}) = 0.
\tag{74}
$$

We have $\kappa(\mu_\epsilon) = \epsilon$ so that $a(\epsilon) = 1/\epsilon$ and thus we see that

$$
f(\mu_\epsilon) = \nu_\epsilon.
\tag{75}
$$

Moreover,

$$
\begin{aligned}
R_{a(\mu_\epsilon)}(\sqrt{\epsilon}) &= R_{1/\epsilon}(\sqrt{\epsilon}) \\
&= \sqrt{\epsilon} + \frac{1}{10}\cos^2\left(\frac{1}{2}\pi\sqrt{\epsilon}\right)\cos\left(\frac{1}{\epsilon}\sqrt{\epsilon}\right) \\
&= \sqrt{\epsilon} + \frac{1}{10}\cos^2\left(\frac{1}{2}\pi\sqrt{\epsilon}\right)\cos\left(\frac{1}{\sqrt{\epsilon}}\right).
\end{aligned}
\tag{76}\tag{77}\tag{78}
$$

Let us assume that there is a continuous map $G : \mathcal{P}([-3,3]) \times [-3,3] \to [-3,3]$, where in the set $\mathcal{P}([-3,3])$ we use the 1-Wasserstein topology such that

$$
f(\mu) = f_G(\mu) = (G(\mu))_{\#}\mu.
\tag{79}
$$

We observe that

$$
f(\mu_\epsilon) = \nu_\epsilon
\tag{80}
$$

implies that when $0 < \epsilon < \frac{1}{2}$ we have $\mu_\epsilon \in \mathcal{M}^+_{fin,dif}([-3,3])$ and

$$
G(\nu_\epsilon, x)|_{x=2} = 2,
\tag{81}
$$

$$
G(\nu_\epsilon, x)|_{x=\sqrt{\epsilon}} = R_{1/\epsilon}(\sqrt{\epsilon}).
\tag{82}
$$

Thus,

$$
\limsup_{\epsilon \to 0+} G(\mu_\epsilon, x)\Big|_{x=\sqrt{\epsilon}} = \limsup_{\epsilon \to 0+} \sqrt{\epsilon} + \frac{1}{10}\cos^2\left(\frac{1}{2}\pi\sqrt{\epsilon}\right)\cos\left(\frac{1}{\sqrt{\epsilon}}\right) = +\frac{1}{10},
\tag{83}
$$

$$
\liminf_{\epsilon \to 0+} G(\mu_\epsilon, x)\Big|_{x=\sqrt{\epsilon}} = \liminf_{\epsilon \to 0+} \sqrt{\epsilon} + \frac{1}{10}\cos^2\left(\frac{1}{2}\pi\sqrt{\epsilon}\right)\cos\left(\frac{1}{\sqrt{\epsilon}}\right) = -\frac{1}{10}.
\tag{84}
$$

Formulas (83), (84), and (73) are in contradiction with the assumption that the map $G : \mathcal{P}([-3,3]) \times [-3,3] \to [-3,3]$ is continuous. This proves the claim. $\square$

The above lemmas yield Proposition 1. □

To discuss the connection of the above counterexample with LLMs, we consider a sequence of tokens $(x_1, x_2, \ldots, x_n) \in \Omega^n$, where $\Omega \subset \mathbb{R}^d$, that are identified with discrete measures $\frac{1}{n} \sum_{i=1}^{n} \delta_{x_i}$ via the map $\iota$ given in formula (4). Below, as an interesting counterexample, we will construct a map $f : \mathcal{M}_{fin}^+([-3,3]) \to \mathcal{M}_{fin}^+([-3,3])$ for which the corresponding map $F = \iota^{-1} \circ f \circ \iota$ maps a sequence of tokens $X = (x_1, x_2, \ldots, x_n) \in [-3,3]^n \subset \mathbb{R}^n$ to a sequence

$$F(X) = (y_1(X, x_1), y_1(X, x_2), \ldots, y_n(X, x_n)) \in [-3,3]^n \subset \mathbb{R}^n.$$

Let us consider an example where $d = 1$ and let $\Omega = [-3,3]$ be the space where we consider the tokes and $B_1 = [-1,1]$ and $B_2 = [-2,2]$ be balls (i.e. intervals) centered at zero.

This map has the following property: Let $n > 1$ be very large and consider a sequence $X_n = (x_1, x_2, \ldots, x_n)$ where

$$x_1, x_2, x_3 \in B_1, \quad x_4, x_5, x_6, x_7, \ldots, x_n \in \Omega \setminus B_2$$

that is, the first three tokens are in the smaller neighborhood of the point 0 and all other tokens are outside the larger neighborhood of the point 0. Denote the image of this sequence of tokens in the map $F$ by

$$F(X_n) = F(x_1, x_2, \ldots, x_n) = (y_1(X_n, x_1), y_2(X, x_2), \ldots, y_n(X, x_n)).$$

When $n$ is large, the measure $\mu_X(B_2)$, of the set $B_2$ with respect to the measure $\mu_X = \frac{1}{n} \sum_{i=1}^{n} \delta_{x_i}$, is small. More precisely, $\mu_X(B_2) = \frac{3}{n}$. Then, when $f$ is the map constructed below in formulas (63)-(65) below, the function

$$x_1 \to (y_1(X_n, x_1), y_2(X_n, x_2), y_3(X_n, x_3))$$

converges to a discontinuous function as $n \to \infty$. This means that when the prompt becomes sufficiently long, then the map $F$ transforms some of the tokens in a possible unstable way. As a possible playful example, two long, almost similar prompts, coded with map which assigns tokens in $\mathbb{R}^d$ for words, so that the names 'Alice' and 'Elise' are mapped to tokens that are very close to each others, that is, $|\iota(Alice) - \iota(Elise)|$ is small. We consider to very long prompts which are the same except their first words (in this example, we use a long 'Lorem ipsum' text that is commonly used in graphic design and publishing as a dummy or placeholder text). The promts

$$X_n = (Alice, is, studying, the, text, Lorem, ipsum, dolor, sit, amet, \ldots, nibh),$$
$$X_n' = (Elise, is, studying, the, text, Lorem, ipsum, dolor, sit, amet, \ldots, nibh),$$

could possibly be mapped in the composition of $F$ and a permutation $S$ (that changes the 1st and the 3rd words)

$$(S \circ F)(X_n) = (The, reader, LOVES, the, text, Lorem, ipsum, dolor, sit, amet, \ldots, nibh),$$
$$(S \circ F)(X_n') = (The, reader, HATES, the, text, Lorem, ipsum, dolor, sit, amet, \ldots, nibh).$$

