# OpenReview forum: "Transformers through the lens of support-preserving maps between measures"
_ICLR.cc/2026/Conference — Submitted to ICLR 2026_

### Official Review · Reviewer_kfm9 · 2025-10-27

**Soundness:** 3
**Presentation:** 1
**Contribution:** 2
**Rating:** 2
**Confidence:** 3

**Summary:**

To analyze the expressivity of transformers with arbitrarily large context, previous work suggested to view the input context as a probability distribution, and extend transformers accordingly to operate on probability distributions. This paper studies mathematically which maps between measures can be realized by a transformer.

**Strengths:**

Possibly, some deep math which I did not understand, to a large extent  because IMHO the writing is poor

**Weaknesses:**

* The author's motivation for addressing this problem is not clear. What do the results teach us about learning theory or expressivity of transformers? Why is this inquiry interesting, given prior work on transformer universality?
* There are lots of abstract mathematical definitions without an explanation of how they relate to the standard formulation of transformers. For example, why are we interested in maps f between measures, and why are we interested in them being generated by 'in context maps'?
* The main result provides a universality guarantee under some condition which is uninuitive. It is not clear how reasonable it is.

**Questions:**

I do not think the paper can be accepted in its present form. If the authors wish to despute this claim, they can try to address the weaknesses mentioned above.

Below are some minor writing suggestions **which should not be addressed in the rebuttal**
* Line 40:" Leverage contexts of arbitrary length" sounds too strong a statement
* Line 41: "the previous work". The word "the" should be omitted.
* Line 50 starting from "the size". This sentence  could be rewritten for clarity.
* Line 105 and elsewhere. I would write "certain" without the parenthesis.
* I was confused by Definition 2. In the definition you mention a "in context map", but you did not define this previously. My understanding is that any $G$ of this form is considered an in context map. Correct? It would be good to clarify this.

---

> ### Author Response · Authors · 2025-11-20
>
> We would like to thank the reviewer for the careful reading and the constructive comments. Below, we address the main concerns.
>
> > The author's motivation for addressing this problem is not clear. What do the results teach us about learning theory or expressivity of transformers? Why is this inquiry interesting, given prior work on transformer universality?
>
> >There are lots of abstract mathematical definitions without an explanation of how they relate to the standard formulation of transformers. For example, why are we interested in maps f between measures, and why are we interested in them being generated by 'in context maps'?
>
> In several prior works, Transformers have been studied from a measure-theoretic perspective, where a Transformer $f$ is interpreted as a map  acting on probability measures. Approximating mappings between measures is highly relevant for many operator-learning tasks, especially in PDEs.  For instance, the solution operator of the Vlasov equation provides a canonical example of a support-preserving map, which is one of the main classes of operators that our theory targets.
> Furthermore, recent prior works show that a Transformer can be written
> in the form
>
> $
> f(\mu) = G(\mu)_{\sharp}\mu,
> $
>
> where $G(\mu):\mathbb{R}^d\to\mathbb{R}^d$ is an in-context map depending
> on the input measure $\mu$.
> In other words, a Transformer acts as the pushforward of $\mu$ under a input-dependent transport map. Any pushforward map $G(\mu)_{\sharp}\mu$ necessarily preserves the support of
> $\mu$, but the converse direction—that every support-preserving map can be represented as such an in-context pushforward—is not obvious a priori. Clarifying this relationship is the motivations of our work.
>
> Based on the above clarification, our novelty lies in establishing the converse direction: we show that every support-preserving map can be represented as a pushforward map induced by an in-context map.  We emphasize that our main contribution is not Corollary 1, but Theorem 1.
>     More precisely, F24 studies the universality of the in-context map $G:(\text{measure},\text{vector})\to\text{vector}$. That is, the approximation ability of $G$ to arbitrary functions of a measure and a point. In contrast, our Theorem 1 addresses a fundamentally different question: it establishes that any support-preserving map $f$ on measures admits a representation of the form
>
> $
> f(\mu)=G(\mu)_{\sharp}\mu
> $
>
> for some in-context map $G(\mu)$. Once this representation theorem is established, F24’s universality result for
> $G$ immediately implies that any support-preserving map can be approximated by a
> measure-theoretic Transformer.
> Thus, Corollary 1 is just a consequence of combining Theorem 1 with the approximation theorem of F24, while Theorem 1 itself constitutes the core
> novel contribution of our work.
>
>
> > The main result provides a universality guarantee under some condition which is uninuitive. It is not clear how reasonable it is.
>
> We would like to clarify that assumptions (B1) and (B2) are in fact the necessary and sufficient conditions for the representation formulas (A1)
> and (A2) of support-preserving maps. Therefore, these assumptions are not arbitrary; rather, they characterize
> exactly when such a representation by an in-context pushforward is possible.
>
> We fully agree that condition (B2) may appear unintuitive.
> This condition arises from a rigorous analysis of the derivative of maps acting on the space of probability measures, where even
> defining derivatives is delicate. Developing a more intuitive interpretation of (B2) is an interesting future direction. The important takeaway is that our results show that mere continuity is insufficient for a support-preserving map to admit the pushforward representation. Some “smoothness’’ with respect to the measure is required.  In Appendix F, we provide counterexamples demonstrating that
> continuity alone fails to guarantee such a representation.
> We believe that this observation is a  meaningful contribution to the ML theory community.

---

> > ### Comment · Reviewer_kfm9 · 2025-11-22
> >
> > I'd like to thank the reviewers for their answers. My opinion remains the same. As someone who is theoretically inclined, but is not familiar with this problem, I was not engaged by the paper. I did not and still do not understand why these mathematical abstractions are of importance to someone interested in transformers for machine learning. I would recommend either submitting to a math journal, or resubmitting to another ML conference after working hard on explaining the relevance of this work to the community.

---

> > > ### Author Response · Authors · 2025-11-24
> > >
> > > > I did not and still do not understand why these mathematical abstractions are of importance to someone interested in transformers for machine learning.
> > >
> > > Thank you very much for your comments. We apologize that the current version did not sufficiently communicate the relevance of our theoretical results to the ML audience. We will revise the introduction to clarify motivation and provide more context.
> > >
> > >
> > > Transformers have recently been applied to **operator learning** for solving PDEs and scientific computing tasks. Several prior works demonstrate that transformers can learn solution operators effectively, for example:
> > >
> > > - Li, Zijie, Kazem Meidani, and Amir Barati Farimani. ``Transformer for partial differential equations' operator learning.'' (2022).
> > >
> > > - Hao, Zhongkai et al. ``GNOT: A General Neural Operator Transformer for Operator Learning.'' (2023).
> > >
> > >
> > > These works motivate the theoretical question of what kinds of operators transformers can fundamentally learn.
> > >
> > > The central contribution of this paper is **Theorem 1**, which provides a representation theorem with necessary and sufficient conditions, for when a transformer can approximate a map between measures. Specifically, we resolve the following question:
> > >
> > > $
> > > \text{What class of operators between measures can transformers learn?}
> > > $
> > >
> > > The answer we obtain is : **support-preserving maps**.
> > > This observation reveals an inherent limitation of transformers as operator learners.
> > > For example, transport-type maps (e.g., Vlasov flow) fall within this class, whereas diffusion-type maps do not, which is in sharp contrast with diffusion-model-based generative methods.

---

> > > > ### Comment · Reviewer_kfm9 · 2025-11-26
> > > >
> > > > I understand these arguments. I slightly raised my score to 4 and reduced my confidence to 2, to reflect that fact that my assessment may be due to ignorance. Nonetheless, I can only speak for myself, and I am not excited about this paper, as explained above.

---

### Official Review · Reviewer_uV5C · 2025-10-29

**Soundness:** 3
**Presentation:** 3
**Contribution:** 3
**Rating:** 6
**Confidence:** 3

**Summary:**

The paper fully characterizes mappings between spaces of positive measures given by the pushforward of a continuous in-context mapping: $\mu \mapsto G(\mu) \mu$. This characterization depends on the regularity of a generalization of the Frechet derivative of the map. As a consequence of prior work that studies the ability of transformers to approximate continuous in-context mappings, the paper deduces that any map satisfying this regularity condition can be approximated by a transformer. As a representative example, the paper demonstrates that the solution map to the Vlasov equation satisfies this regularity condition, and hence can be represented by a transformer. Conversely, the paper also shows that mean-field transformers can also be identified with Vlasov equations in some sense.

**Strengths:**

The interpretation of transformers are measure-to-measure mappings has become important in understanding their ability to solve scientific problems (such as sampling and PDEs), and it is therefore natural to ask which such mappings can be realized by a transformer. This paper takes a first cut at this problem and establishes a necessary and sufficient criterion. I feel that the work has potential for impact, as it gives a recipe for proving approximation results for transformers beyond the Vlasov equation.

In addition, the paper is well-written, and the authors do a nice job of conveying intuition despite the technical nature of the results.

**Weaknesses:**

The main weakness is that the approximation results are purely qualitative and do not provide bounds on the size of the transformer needed to approximate a given map. The authors acknowledge this limitation in the paper.

Additionally, the regularity condition is fairly technical, and thus it is not clear to me how difficult it is to verify it beyond the Vlasov equation, or how abundant are the mappings which satisfy it.

**Questions:**

If I understand correctly, the support-preserving condition is not satisfied by autoregressive/causally-masked transformers, where the prediction of the 'i'-th token depends only on the first 'i'. tokens. I know that this is not the focus of the paper, but it may be worth to comment on this.

---

> ### Author Response · Authors · 2025-11-20
>
> We would like to thank the reviewer for the careful reading and the constructive comments. Below, we address the main concern.
>
>
>
> >Additionally, the regularity condition is fairly technical, and thus it is not clear to me how difficult it is to verify it beyond the Vlasov equation, or how abundant are the mappings which satisfy it.
>
> We would like to clarify that assumptions (B1) and (B2) are in fact the necessary and sufficient conditions for the representation formulas (A1)
> and (A2) of support-preserving maps. Therefore, these assumptions are not arbitrary; rather, they characterize
> exactly when such a representation by an in-context pushforward is possible.
>
> We fully agree that condition (B2) may appear unintuitive.
> This condition arises from a rigorous analysis of the derivative of maps acting on the space of probability measures, where even
> defining derivatives is delicate. Developing a more intuitive interpretation of (B2) is an interesting future direction. The important takeaway is that our results show that mere continuity is insufficient for a support-preserving map to admit the pushforward representation. Some “smoothness’’ with respect to the measure is required.  In Appendix F, we provide counterexamples demonstrating that
> continuity alone fails to guarantee such a representation.
> We believe that this observation is a  meaningful contribution to the ML theory community.

---

> > ### Comment · Reviewer_uV5C · 2025-11-20
> >
> > Thanks for the reply. I understand that the assumptions (B1) and (B2) are necessary and sufficient for the representation formulas of support-preserving maps. I think the result is a meaningful and relevant contribution to the ML community and maintain my score.

---

### Official Review · Reviewer_gZyY · 2025-11-01

**Soundness:** 2
**Presentation:** 1
**Contribution:** 2
**Rating:** 2
**Confidence:** 3

**Summary:**

# Context
For context, my original recommendation was to desk reject the submission due to breaching author anonymity, due to the reasons provided below in my original review. After the review deadline, the AC told me that they had consulted with the senior AC and PCs, who disagreed with me regarding author anonymity. I maintain my original stance, but the AC has asked me to write an updated review after the review deadline, due the SAC/PC position. I have done so (on the weekend at short notice), as detailed below, but have kept my unedited original review for posterity. I have done my best to write my updated review independently of, and unbiased by, my original reading of the work. I strongly urge the authors to remove the first person "we" in the abstract in the next update of their work.

======================================

# Original Review, Nov 1

Recommending desk reject as the authors appear to have breached anonymity in my reading.

Namely, the authors write on line 13 of the abstract: "In previous work, **we** studied the ability of these architectures to handle an arbitrarily large number of context tokens", and on line 50 write "The size of this context might be very long, possibly arbitrarily long, which has been addressed in Furuya et al. (2024) that concerns the transformers as universal in-context learners."

As per the author guide, https://iclr.cc/Conferences/2026/AuthorGuide, "Any paper where author identity is revealed in either the main text or the supplementary material will be desk rejected. Note that related arxiv papers by the same authors do not break anonymity; if cited, these should be cited in third person.". The use of "we" in line 13 is first person.

===========================

# Updated review, Nov 8
This submission studies "transformers through the lens of support-preserving maps between measures". Namely, the authors build on previous work, Furuya et al, 2024 (F24), that establishes the universality of transformers within the family of continuous in-context mappings (definition 2). This work builds on F24 by characterising the nature of "support-preserving maps" (definition 1) that transformers can approximate, by identifying a certain smoothness condition (theorem 1, B2) that is essential for such in-context maps support-preserving maps to be in-context maps. This essentially covers the first 8 pages of the submission in my reading. The last page of the main paper is a corollary of theorem 1, that shows that the solution of the Vlasov equation satisfies the introduced conditions.

I am giving the submission a score of 2, for reasons detailed below, but would give a 3 if there was the option.

**Strengths:**

- The paper appears to be correct and demonstrates a condition to characterise the family of in-context maps within the family of support-preserving maps.
- There is a significant amount of mathematical maturity that has gone into this paper (see also weaknesses).

**Weaknesses:**

Points 1-3 are all related and combine for my main criticism of the submission.
1. **Technical novelty** The paper is heavily inspired by F24, and does not appear to be a significant technical advancement upon F24. I am not an expert here, but as I understand it, the technical contribution of this paper is to identify conditions (in theorem 1) that separate/equate the family of in-context maps from the family of support-preserving maps. Then this paper leans on F24 in Corollary 1, which has already established the universality of transformers within the family of in-context maps, to show the universality of transformers for support-preserving maps that satisfy conditions B1/B2 as a corollary. Based on this reading, it appears like F24 provided a significant result, and that this submission is a more minor addition on top of F24.
2. **Technical relevance to ICLR community** Relatedly, if on the other hand the results of Theorem 1 (providing the conditions for equivalence between "in-context maps" and "support-preserving maps") are significant technically, it doesn't feel like an ML conference like ICLR is the best venue to appreciate the importance of the specific smoothness conditions (B2) that separate two families of maps between measures. It would be helpful if the authors could better motivate what this distinction means, i.e. what does it mean to satisfy B2 or not, and which bigger picture intuition can someone working on transformers take away from this?
3. **Motivation** Also relatedly, the authors do not appear to do a good job of motivating the significance/importance of their work, in my reading. For example, with "support-preserving maps" the authors write they are "natural" to study transformers on line 170 and provide some links, but the context should be provided on line 170 and, in any case, the links provided do not make the motivation sufficiently clear to this reviewer. Given the importance of "support-preserving maps" for the technical contribution of this submission (aforementioned in point 1), this motivation needs to be stronger. Indeed, "support-preserving maps" appears to be a term coined in this submission, as opposed to a standard term in the literature. To play devil's advocate, what does the knowledge that certain support-preserving maps are in-context maps add for a transformers/deep learning researcher, on top of the knowledge that transformers are universal within the family of in-context maps (from F24)? Likewise, what is the importance of the finding that Vlasov equation solutions satisfy certain conditions in Section 4.2 and can be approximated by a transformer? The context for this is somewhat lacking (I hadn't heard of Vlasov equations before and I imagine most of the ICLR community would not have either).
4. **Lack of accessibility** As mentioned, the submission requires quite heavy mathematical maturity to parse. This by itself is not a weakness, but I think the way the paper is written has several drawbacks for enabling a reader to understand it. One drawback is that the authors do not provide sufficient context of the mathematical objects studied in the paper, for the reader: for example the lacking motivation for "support-preserving maps" or Vlasov flows, as mentioned above. But in addition, the reader is expected to know a lot of notation/nomenclature that is often not found in ICLR papers, e.g. zero-homogeneous extensions or uniformly continuous or even basic measure theory (borel sets, sigma algebras etc), for which it would be helpful to have definitions or references provided. There are also a few typos/omissions which appear to be oversights that compromise the reader accessibility. For example, $\cal{M}$ missing $\cal{M}^{+}$ on line 143, or $\cal{P}_{fin}$ is not defined on line 173 (this is bad as it's not clear that $f$ on line 180 is well defined as a result), or on line 192 what is $i$ (should it be $\forall i$?)
5. **Quantitative results** The authors mention the lack of quantitative results but it would for sure improve the broader relevance of the paper to the ICLR community to know how the universality is affect by e.g. changing the number of layers or the number of heads? Likewise, what is special about transformers to enable universality as opposed to other sequence models, e.g. state space models, which are known to perform worse on in context learning (though I guess this is outside the scope of the paper at present).

**Questions:**

- Why is the background material on basic transformers introduced in line 188, in the middle of an unrelated section that is describing Wasserstein metrics/distances?

**Details Of Ethics Concerns:**

author anonymity (this is from my original review, and I cannot delete in the updated review).

---

> ### Author Response · Authors · 2025-11-20
>
> We would like to thank the reviewer for the careful reading and the constructive comments. First of all, we sincerely apologize for the anonymity issue. We have already corrected it.
>
> Below, we address the main concerns.
>
> > 1. Technical novelty & 3. Motivation
>
> In several prior works, Transformers have been studied from a measure-theoretic perspective, where a Transformer $f$ is interpreted as a map  acting on probability measures. Approximating mappings between measures is highly relevant for many operator-learning tasks, especially in PDEs.  For instance, the solution operator of the Vlasov equation provides a canonical example of a support-preserving map, which is one of the main classes of operators that our theory targets.
> Furthermore, recent prior works show that a Transformer can be written
> in the form
>
> $
> f(\mu) = G(\mu)_{\sharp}\mu,
> $
>
> where $G(\mu):\mathbb{R}^d\to\mathbb{R}^d$ is an in-context map depending
> on the input measure $\mu$.
> In other words, a Transformer acts as the pushforward of $\mu$ under a input-dependent transport map. Any pushforward map $G(\mu)_{\sharp}\mu$ necessarily preserves the support of
> $\mu$, but the converse direction—that every support-preserving map can be represented as such an in-context pushforward—is not obvious a priori. Clarifying this relationship is the motivations of our work.
>
> Based on the above clarification, our novelty lies in establishing the converse direction: we show that every support-preserving map can be represented as a pushforward map induced by an in-context map.  We emphasize that our main contribution is not Corollary 1, but Theorem 1.
>     More precisely, F24 studies the universality of the in-context map $G:(\text{measure},\text{vector})\to\text{vector}$. That is, the approximation ability of $G$ to arbitrary functions of a measure and a point. In contrast, our Theorem 1 addresses a fundamentally different question: it establishes that any support-preserving map $f$ on measures admits a representation of the form
>
> $
> f(\mu)=G(\mu)_{\sharp}\mu
> $
>
> for some in-context map $G(\mu)$. Once this representation theorem is established, F24’s universality result for
> $G$ immediately implies that any support-preserving map can be approximated by a
> measure-theoretic Transformer.
> Thus, Corollary 1 is just a consequence of combining Theorem 1 with the approximation theorem of F24, while Theorem 1 itself constitutes the core
> novel contribution of our work.
>
>
> >2. Technical relevance to ICLR community
>
> We would like to clarify that assumptions (B1) and (B2) are in fact the necessary and sufficient conditions for the representation formulas (A1)
> and (A2) of support-preserving maps. Therefore, these assumptions are not arbitrary; rather, they characterize
> exactly when such a representation by an in-context pushforward is possible.
>
> We fully agree that condition (B2) may appear unintuitive.
> This condition arises from a rigorous analysis of the derivative of maps acting on the space of probability measures, where even
> defining derivatives is delicate. Developing a more intuitive interpretation of (B2) is an interesting future direction. The important takeaway is that our results show that mere continuity is insufficient for a support-preserving map to admit the pushforward representation. Some “smoothness’’ with respect to the measure is required.  In Appendix F, we provide counterexamples demonstrating that
> continuity alone fails to guarantee such a representation.
> We believe that this observation is a  meaningful contribution to the ML theory community.

---

> > ### Comment · Reviewer_gZyY · 2025-11-22
> > **Maintaining my score**
> >
> > Thanks for the reply. The author's response does not significantly update me on my main concerns: I understood that theorem 1 was the main result of this submission in my initial reading, but my concern "it appears like F24 provided a significant result, and that this submission is a more minor addition on top of F24." remains. The authors have also not responded to several of my other concerns, particularly on writing/accessibility for reader. I have read the other reviews, noting that some concerns are shared with kfm9, but uV5C is more positive. I maintain my score.

---

> > > ### Author Response · Authors · 2025-11-24
> > >
> > > Thank you very much for your comments.
> > >
> > > First, we would like to clarify that **Theorem 1 is independent of F24 and cannot be derived using the ideas from F24**.
> > > The work F24 analyzes only in-context maps of the form
> > >
> > > $
> > > (\text{measure}, \text{vector}) \longrightarrow \text{vector},
> > > $
> > >
> > > which does **not** cover the class of mappings
> > >
> > > $
> > > \text{measure} \longrightarrow \text{measure}.
> > > $
> > >
> > > In contrast, **Theorem 1 provides a representation theorem together with necessary and sufficient conditions** for when a transformer can approximate a map between measures.
> > > This resolves the fundamental question:
> > >
> > > $
> > > \text{What class of operators between measures can transformers learn?}
> > > $
> > >
> > > The answer we obtain is: **support-preserving maps**.
> > > This observation reveals an inherent limitation of transformers as operator learners.
> > > For example, transport-type maps fall within this class, whereas diffusion-type maps do not, which is in sharp contrast with diffusion-model-based generative methods.
> > >
> > > Regarding your remaining concern:
> > >
> > > ``it appears like F24 provided a significant result, and that this submission is a more minor addition on top of F24.''
> > >
> > > If our response does not resolve your concern, we would appreciate it if you could clarify **in what sense** the result presented here appears to be a minor extension.
> > >
> > >
> > >
> > >
> > > >Lack of accessibility
> > >
> > > Thank you very much for pointing out the accessibility concerns.
> > > We sincerely apologize for the lack of accessibility in the current version.
> > > We will revise the manuscript to include definitions, references, and explanations for key concepts and notations
> > > (e.g., measure-theoretic terminology, zero-homogeneous extensions, uniform continuity, etc.).
> > >
> > >
> > > **Regarding the motivation for support-preserving maps and Vlasov flows:**
> > > We believe that transformers, when viewed as mappings between probability measures, can play an important role in operator learning for PDEs.
> > > There are many recent works applying transformers to learn solution operators of PDEs, for example:
> > >
> > > - Li, Zijie, Kazem Meidani, and Amir Barati Farimani.
> > >     ``Transformer for partial differential equations' operator learning.'' (2022).
> > > - Hao, Zhongkai et al.
> > >     ``GNOT: A General Neural Operator Transformer for Operator Learning.'' (2023).
> > >
> > >
> > > As mentioned in our previous response, our result highlights an intrinsic limitation:
> > > transformers can approximate only **support-preserving** maps between measures.
> > > A representative positive example is the Vlasov flow (a nonlinear transport map), whereas PDEs involving diffusion terms generally fall outside this class. In the revised manuscript, we will discuss these views for operator learning.
> > >
> > >
> > > >Quantitative results
> > >
> > > Thank you for the insightful comment. We agree that quantitative results would be valuable for the broader community. However, we would like to emphasize that the main contribution of this paper is not the universality result, but rather the **representation theorem** characterizing support-preserving maps via push-forward operators. In this sense, extending the current analysis to quantitative bounds (e.g., dependence on depth or the number of heads) is beyond the scope of this work.

---

### Meta-Review · Area_Chair_QP5j · 2026-01-02

**Summary:**

The authors model transformers as a map between input and output tokens in terms of measures. This is achieved by allowing for the permutation invariance of input tokens (i.e. shuffling the order of words), which yields a measure representation. The authors provide if and only if conditions for universal representation of measure flow maps, and connect the transformer to Vlasov equations (mean field transport equations).

Overall, the strengths found by the reviewers are towards the directions of mathematical precision and connection to the Vlasov equation.

The weaknesses found by the reviewers were several fold:
- The paper was overly technical and potentially difficult to read
- It is unclear what the impact of this work yields
- The results are incremental compared to previous work

Personally, the technicality of this work does not bother me, and I believe the characterization results are potentially interesting in a mathematical context, so I don't believe these are issues. However, I do believe the latter two criticisms are valid.

Subjectively, I would also like to add several of my own criticisms of this work.

Firstly, impact is significantly diminished by the modelling choice of allowing for token permutation invariance, which no longer allows the modelling of transformers as sequence to sequence maps. In particular, we lose the causal structure completely. This definitely bothers me as in all applications of the transformer, whether it's NLP or vision, the ordering of the tokens plays a key role.

Secondly, I would also add that expressivity results are not directly relevant to training. Surely most people today would believe that transformers can overfit on any sequence patterns, which implies that the architecture do not lack expressivity. This would certainly raise the concern of what kind of impact expressivity results have on deep learning today.

Thirdly, I would also add that the link with Vlasov equations is somewhat trivial. Not only have it been discussed previously by other works on transformers, but it fundamentally arises in all situations when you allow for permutation invariance. This is not particularly mathematically novel in my opinion.

To put these thoughts together, I believe the key issue is that the paper contributes a set of precise mathematical results that will be difficult to find impact in ML today, which in my opinion is still the main purpose of the ICLR conference. Therefore I will still recommend reject.

**Reviewer Concerns:**

I believe the issues of impact and incremental improvement over the previous work were not adequately addressed. This is really the key issue here.

**Reviewer Scores:**

gZyY - 2 (remain the same)
kfm9 - 4 (raised as commented)
uV5C - 6 (remain the same)

---

### Decision · Program_Chairs · 2026-01-26

Reject